*Method*

EMBO
Molecular Medicine

# High-purity AAV vector production utilizing recombination-dependent minicircle formation and genetic coupling

Hao Liu [1], Nan Liu [1], Chen Zhou[1], Ailing Du [1], Mayank Kapadia [1], Phillip W L Tai [1], Erik Barton[2], Guangping Gao [1,3]✉ & Dan Wang [1,4]✉

## Abstract

Triple transfection of HEK293 cells is the most widely used method for producing recombinant adeno-associated virus (rAAV), a leading gene delivery vector for human gene therapy. Despite its tremendous success, this approach generates several vector-related impurities that could potentially compromise the safety and potency of rAAV. In this study, we introduce a method for high-purity AAV vector production utilizing recombination-dependent minicircle formation and genetic coupling (AAVPure^Mfg). Compared with traditional triple transfection, AAVPure^Mfg substantially improves vector purity by reducing prokaryotic DNA contaminants by 10- to 50-fold and increasing the full capsid ratio up to threefold. Mechanistically, Bxb1-mediated excision of the transgene cassette generates a minicircle cis construct devoid of bacterial sequences and ensures synchronized colocalization of trans and cis constructs in productive cells. Furthermore, we developed iterations that enhance vector genome homogeneity and streamline the production of rAAV with various transgenes, serotypes, and ITR configurations. Overall, our findings demonstrate that AAVPure^Mfg overcomes the inherent limitations associated with triple transfection, offering a broadly applicable and easy-to-implement method for producing high-purity rAAV with reduced plasmid costs.

**Keywords** AAV Manufacturing; Vector Impurities; Empty Capsid; Plasmid Backbone; AAVPure^Mfg
**Subject Categories** Genetics, Gene Therapy & Genetic Disease; Methods & Resources

## Introduction

Recombinant adeno-associated virus (rAAV) has emerged as the leading viral vector for in vivo gene therapy delivery due to its low immunogenicity, long-term episomal persistence and transgene expression, and engineering potential for tissue-specific targeting (Kuzmin et al, 2021; Wang et al, 2019; Wang et al, 2024). With seven US FDA-approved rAAV medicines and hundreds of ongoing clinical trials for rare and common diseases, manufacturing high-purity rAAV to ensure vector safety and potency has become increasingly critical (Srivastava et al, 2021). Since the advent of triple transfection (Xiao et al, 1998), it has been the predominant method for rAAV production. Triple transfection involves three plasmids co-transfected into HEK293 cells at a roughly equal molar or mass ratio: a helper plasmid (pHelper) providing adenovirus helper genes, a trans-complementing packaging plasmid (pTrans) that expresses AAV *Rep* and *Cap* genes, and a cis plasmid (pCis) that contains a therapeutic transgene cassette flanked by AAV inverted terminal repeats (ITRs) (Fig. 1A). Despite being widely used to produce rAAV at all scales, triple transfection-based processes result in various vector-related impurities, such as prokaryotic plasmid backbone DNA encapsidation and high levels of empty capsids at harvest.

rAAV produced by triple transfection typically contains 1%–10% plasmid backbone DNA (relative to transgene DNA) that is mostly derived from pCis; some reports indicate levels as high as 26.1% (Brimble et al, 2023; Hauck et al, 2009; Schnodt et al, 2016; Wright, 2014). These prokaryotic sequences often include highly immunogenic CpG motifs and potentially harmful open reading frames (ORFs), such as antibiotic resistance genes. We and others have shown that the packaged prokaryotic DNA persists in animal tissues following rAAV administration in mice, dogs, and non-human primates (Chadeuf et al, 2005; Liu et al, 2024). These prokaryotic DNA, together with their undesired RNA and/or protein products, may trigger immune responses and cause cytostatic effects in recipients (Liu et al, 2024; Schnodt and Buning, 2017; Wright, 2014). Indeed, as little as 0.87% of plasmid backbone in rAAV was shown to cause inflammation and toxicity in

[1]Department of Genetic and Cellular Medicine, University of Massachusetts Chan Medical School, Worcester, MA 01605, USA. [2]Pfizer Inc., Worldwide Research, Development and Medical, Bioprocess Research and Development, Chesterfield, MO 63017, USA. [3]Department of Microbiology, University of Massachusetts Chan Medical School, Worcester, MA 01605, USA. [4]RNA Therapeutics Institute, University of Massachusetts Chan Medical School, Worcester, MA 01605, USA. ✉E-mail: Guangping.Gao@umassmed.edu; Dan.Wang@umassmed.edu

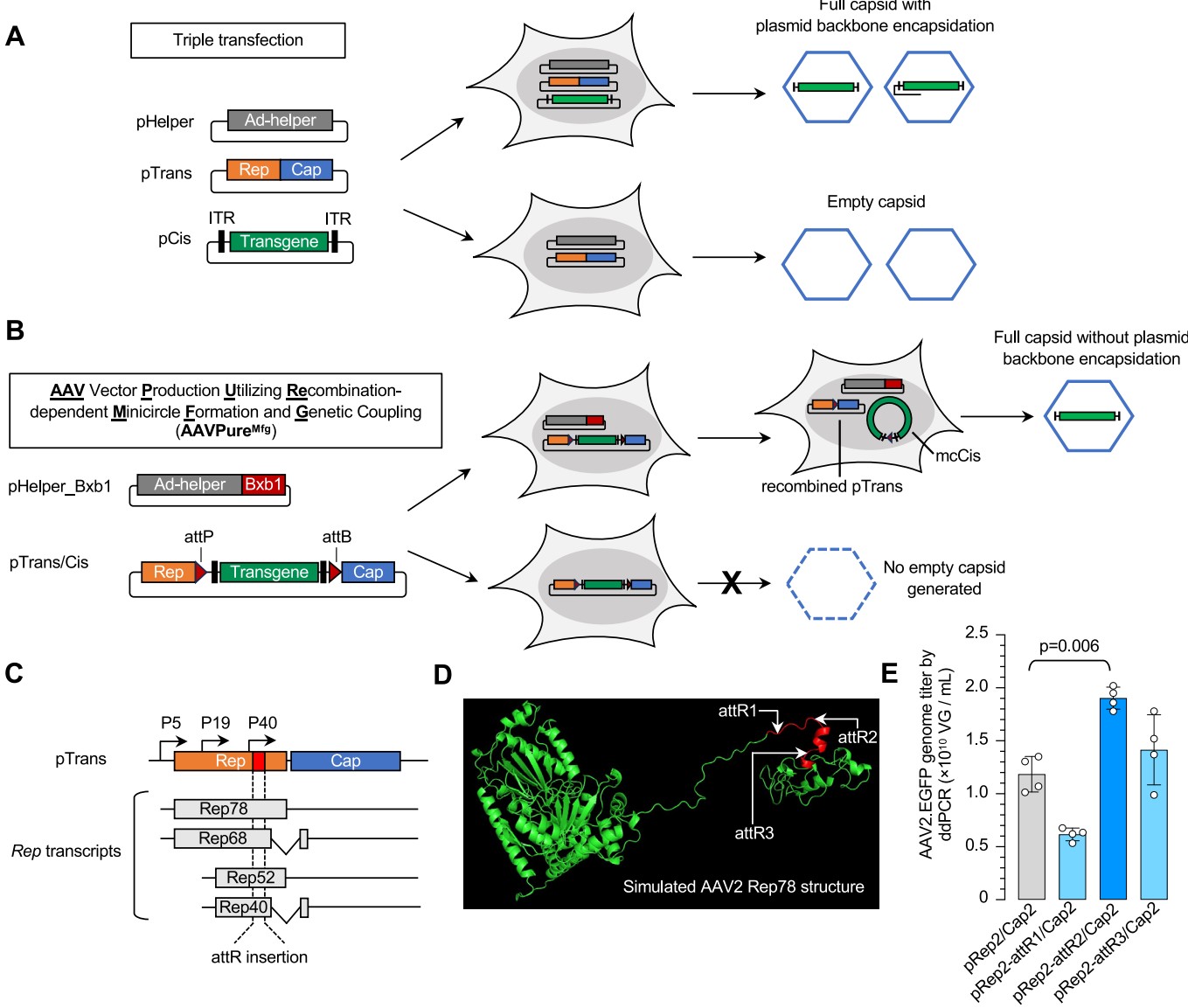

**Figure 1. Schematics comparing triple transfection and AAVPure^Mfg.**

(A) Schematic diagram illustrating how rAAV impurities, including plasmid backbone encapsidation and empty capsid, are generated in triple transfection. After co-transfection, when the three plasmids—pHelper, pTrans, and pCis—are co-localized within the same nuclei of HEK293 cells, Rep rescues the ITR-flanked transgene from the pCis plasmid, generating both the desired full-length rAAV genome and the undesired backbone-containing rAAV genome, both of which are encapsidated. Empty capsids are generated when only pHelper and pTrans coexist in the nuclei of HEK293 cells. (B) Schematic diagram illustrating how rAAV impurities, including plasmid backbone encapsidation and empty capsid, are mitigated in AAVPure^Mfg. After co-transfection, when pHelper_Bxb1 and pTrans/Cis are co-localized in the same nuclei of HEK293 cells, Bxb1 catalyzes the recombination of attP/attB-flanked transgene cassette, generating minicircle Cis construct (mcCis) that contains no prokaryotic backbone sequence, and the reconstituted pTrans with intact Rep and Cap function. Because mcCis does not contain plasmid backbone, cis construct replication only generates desired full-length rAAV genome, thus avoiding plasmid backbone encapsidation. pTrans/Cis itself cannot generate an empty capsid, because the inserted ITR-flanked transgene cassette prematurely terminates Cap gene expression. (C) The gene structure of pTrans plasmid and the expressed Rep transcripts. attR is chosen to be inserted between P40 and intron, located at the 3′ region of Rep gene as shown in red. (D) Simulated AAV2 Rep78 protein structure, with the region between P40 and the intron highlighted in red. The attR insertion sites are indicated by white arrows. (E) Packaging yield of AAV2.EGFP using unmodified pTrans (pRep2/Cap2) or modified variants (pRep2-attR/Cap2). Small-scale rAAV production was conducted in 12-well plate transfected with an equal amount of pRep2/Cap2 or pRep2-attR/Cap2 variants. Crude lysate was harvested 72 h post transfection followed by three successive freeze–thaw cycles. Cleared crude lysates after centrifugation were treated with DNase-I and proteinase K, followed by droplet digital PCR (ddPCR) to determine the genome titer. In (E), data are mean ± s.d. of biological replicates, $n = 4$ for each group. Statistical analysis was performed using one-way ANOVA followed by Dunnett's multiple comparisons test against the pTrans group. The exact $P$ value was indicated in the figure. Source data are available online for this figure.

non-human primates (Keiser et al, 2021; Taylor et al, 2024). In clinical trials, plasmid backbone DNA contaminants in rAAV were considered as a potential contributor to adverse events and led to clinical holds by FDA (Pena, 2018; Shen et al, 2022).

Because encapsidated prokaryotic DNA cannot be eliminated by downstream processing, mitigation strategies mainly focus on plasmid design and upstream processing. For instance, using pCis with an oversized backbone that exceeds AAV packaging capacity has been shown to decrease plasmid backbone encapsidation by approximately fivefold (Hauck et al, 2009; Wright, 2014); including transcriptional insulators adjacent to ITRs further suppresses backbone DNA expression in transduced cells (Taylor et al, 2024). We recently showed that simply reducing pCis usage in triple transfection can lower plasmid backbone encapsidation by two- to seven-fold (Liu et al, 2024). However, these strategies only lead to modest improvement. Replacing plasmid with minicircle DNA or doggybone DNA that lacks plasmid backbone sequences proves to more effectively reduce prokaryotic DNA encapsidation (Kinga Karbowniczek et al, 2017; Schnodt et al, 2016), but their implementation is hindered by technical complexities associated with DNA manufacturing and high synthesis error rates (Pupo et al, 2022).

In addition to prokaryotic DNA contaminants, empty capsids are also a major rAAV impurity, representing 70–90% of total rAAV particles in crude harvest (Schnodt and Buning, 2017; Wright, 2014). Several studies showed that excessive empty capsids could compromise vector transduction efficiency by competitive binding with cell surface receptors (Parker et al, 2003; Gao et al, 2014). Furthermore, empty capsids can trigger or exacerbate capsid-directed immune responses, resulting in the clearance of transduced cells and further compromising therapeutic efficacy (Hosel et al, 2012; Manno et al, 2006; Mingozzi et al, 2007; Nathwani et al, 2014; Schnodt and Buning, 2017; Wright, 2014). Although advances in downstream processing have improved full capsid ratio in general, the optimal process usually needs to be individually developed for specific rAAV products, and the success highly depends on the full capsid ratio in the starting material (Joshi et al, 2021; Qu et al, 2007; Roach et al, 2024).

In this study, we introduce AAVPure^Mfg, a method for high-purity AAV vector production utilizing recombination-dependent minicircle formation and genetic coupling. AAVPure^Mfg simultaneously reduces plasmid backbone encapsidation to unprecedentedly low levels in plasmid-based rAAV manufacturing, while also improving full capsid ratio at harvest to facilitate downstream purification. Furthermore, we demonstrate the molecular mechanism underlying empty capsid formation in standard triple transfection and how AAVPure^Mfg overcomes this limitation.

## Results

### Design and characterization of AAVPure^Mfg components

We hypothesized that empty capsids can be generated due to unbalanced uptake of pTrans and pCis during triple transfection in HEK293 cells. Specifically, functional expression of Rep and Cap from pTrans in the absence of pCis is a major source of empty capsid formation (Fig. 1A). Therefore, we designed AAVPure^Mfg, aiming to ensure the coexistence of the trans and cis constructs in

the same nucleus following transfection. It comprises two plasmids (Fig. 1B): pHelper-Bxb1 that delivers adenoviral helper genes and the recombinase Bxb1, and pTrans/Cis with an attP/attB-flanked cis construct inserted into the 3' region of Rep gene in the standard pTrans (Jusiak et al, 2019). Upon co-transfection into HEK293 cells, Bxb1-mediated attP/attB recombination reconstitutes pTrans and generates a minicircle Cis construct (mcCis) devoid of prokaryotic DNA sequence; the recombined pTrans affords Rep and Cap expression, and the mcCis contains ITR-flanked transgene, serving as the replication template to generate high-purity vector genomes without plasmid backbone DNA. Importantly, when HEK293 cells receive only pTrans/Cis, the ITR-flanked transgene cassette embedded in the Rep gene serves as a disrupting insertion that abolishes Cap expression, thus preventing empty capsid formation (Fig. 1B). The success of AAVPure^Mfg hinges on two key gene expression control events: (1) when Bxb1-mediated recombination occurs, the reconstituted pTrans should enable functional Rep and Cap expression for vector packaging, and (2) in the absence of Bxb1 (i.e., no recombination), the intervening cis construct must block Cap expression to prevent empty capsid formation.

To achieve the first control event, one challenge is that attP/attB recombination leaves an attR sequence within the Rep open reading frame (ORF), potentially compromising Rep and/or Cap expression. Therefore, we set out to insert an in-frame attR site between the P40 promoter (driving Cap transcription) and the intron (Fig. 1C). This insertion region was strategically chosen for two reasons: (1) following attP/attB recombination, although the resulting attR sequence is transcribed and included in the Cap transcripts, it does not disrupt intron integrity and maintains normal Cap mRNA splicing that is essential to generate capsid protein isoforms (i.e., VP1, VP2, VP3) (Trempe and Carter, 1988), and (2) this insertion region in the Rep protein is predicted to be disordered (Fig. 1D) and has been shown to be amenable to mutagenesis, suggesting it may tolerate the additional residues encoded by attR (Baek et al, 2021; Jain et al, 2024).

We modified the pTrans for packaging AAV2 serotype vector (pRep2/Cap2) by inserting attR at three positions individually within the chosen region (pRep2-attR1/Cap2, pRep2-attR2/Cap2, pRep2-attR3/Cap2). As expected, all three pRep2-attR/Cap2 variants afforded rAAV production following a standard triple transfection scheme (Fig. 1D,E), without altering the VP isoforms ratio or abundance as determined by western blot analysis (Appendix Fig. S1A,B). Consistent with in-frame attR insertion, Rep-attR fusion proteins showed a slight molecular weight shift compared with the wild-type Rep expressed from unmodified pTrans (Appendix Fig. S1B, bottom panel). When an equal volume of rAAV-containing cell lysates was used in an in vitro transduction assay (Appendix Fig. S1A), reporter transgene expression well correlated with rAAV titers (compare Appendix Fig. S1C with Fig. 1E), consistently showing that pRep2-attR2/Cap2 slightly outperformed the other two variants. Therefore, we focused on the attR2 insertion position in the following studies.

To achieve the second control event, we replaced attR2 with the attP/attB-flanked cis construct (i.e., attP-ITR-CB6-EGFP-pA-ITR-attB) in the same orientation as Rep (Appendix Fig. S2A), so that the polyadenylation signal (pA) within the transgene cassette could prematurely terminate Cap transcription and blunt VP expression (Appendix Fig. S2B, top panel, lane 5). We noticed that Rep protein

expression from pTrans/Cis was also abolished (Appendix Fig. S2B, bottom panel, lane 5). To investigate whether pTrans/Cis could express a residual amount of functional Rep variants that were not detectable by western blot, we devised a Rep-dependent vector genome amplification assay (Appendix Fig. S2C). In this assay, a small amount of pCis (pAAV.mCherry) was co-transfected with pHelper and pRep (no Cap), so that the vector genome (i.e., ITR-flanked mCherry cassette) was replicated, resulting in increased mCherry expression (Appendix Fig. S2C, compare panel 2 and panel 3). In contrast, replacing pRep (no Cap) with pTrans/Cis failed to amplify mCherry fluorescence signal (Appendix Fig. S2C, panel 4), demonstrating successful silencing of *Rep* in pTrans/Cis.

To deliver the *Bxb1* gene, we opted to insert it downstream of the adenoviral DNA-binding protein (DBP) coding sequence in pHelper via either a T2A ribosomal skipping element (pHelper_DBP-2A-Bxb1) or an internal ribosomal entry site (IRES; pHelper_DBP-IRES-Bxb1) (Appendix Fig. S3A). Both designs leverage the endogenous adenoviral *DBP* gene promoter and 3' UTR, so that the entire *Bxb1* expression cassette (from *DBP* promoter to 3'UTR) spans ~7 kb and greatly exceeds the AAV packaging capacity (5 kb), thus preventing the potential packaging of a functional *Bxb1* gene in rAAV products. In a Bxb1-dependent reporter assay, both designs successfully mediated attP/attB recombination to reconstitute reporter (i.e., enhanced green fluorescent protein, EGFP) expression (Appendix Fig. S3B), demonstrating functional Bxb1 expression.

## AAVPure[Mfg] improves AAV vector purity

We designated the T2A and IRES designs as AAVPure[Mfg] 1.0 and AAVPure[Mfg] 1.1, respectively, and initially produced AAV2.EGFP vectors for characterization (Fig. 2A,B). Compared with triple transfection, both AAVPure[Mfg] 1.0 and 1.1 produced similar rAAV genome titers using ~20% less plasmid by mass (Fig. 2C; Table EV1). The resulting AAV2.EGFP vectors transduced HEK293 cells with similar efficiency (Fig. 2D). Notably, despite comparable genome titers, the capsid titers of both AAVPure[Mfg] vectors were reduced by half (Fig. 2E). Consequently, the full capsid ratio—calculated by normalizing the genome titer to capsid titer—increased approximately twofold from 10.8% in triple transfection to 23.3% with AAVPure[Mfg] 1.0 and 21.6% with AAVPure[Mfg] 1.1 (Fig. 2F).

Consistent with previous reports that synthetic minicircle DNA significantly reduces plasmid backbone contamination (Schnodt et al, 2016), the abundance of encapsidated prokaryotic ampicillin resistance gene (*AmpR*) in rAAV products was diminished by 38-fold, from 5.2% in triple transfection to 0.14% in AAVPure[Mfg] 1.0. Similarly, 0.19% of *AmpR* was observed in AAVPure[Mfg] 1.1, representing a 28-fold reduction (Fig. 2G). Thus, AAVPure[Mfg] 1.0 exhibited slightly lower *AmpR* contamination than AAVPure[Mfg] 1.1, although the difference was not statistically significant. We chose to focus on pHelper_DBP-2A-Bxb1 for further characterization, which demonstrated its ability to restore Rep and Cap expression from pTrans/Cis (Appendix Fig. S2B, lane 4), enabling ITR-flanked vector genome amplification (Appendix Fig. S2C, panel 5).

To assess the broad applicability of AAVPure[Mfg] 1.0, we tested this platform to produce AAV9 and AAV8 vectors, two serotypes commonly used in clinical development (Issa et al, 2023; Kuzmin et al, 2021; Wang et al, 2019) (Fig. EV1A). Consistent with our findings with AAV2, AAVPure[Mfg] 1.0 produced similar rAAV9 and

rAAV8 genome titers as compared with triple transfection, whereas the full capsid ratios increased by two- to threefold with markedly lower *AmpR* gene encapsidation (Fig. EV1B,C). To comprehensively characterize DNA impurities, we produced more AAV9 vectors and subjected the packaged DNA to single-molecule, long-read DNA sequencing (Tai et al, 2018) (Appendix Fig. S4A,B). The plasmid backbone DNA contamination in AAVPure[Mfg] 1.0 was reduced by 30-fold (Table EV2), consistent with the ddPCR results (Appendix Fig. S4C). DNA impurities from other sources, including the host cell genome and pHelper, were comparable. (Table EV2). No replication-competent AAV (rcAAV) genome (i.e., ITR-Rep-Cap-ITR) was detected, despite the presence of one copy of Rep-Cap and low levels of ITR-Rep and Cap-ITR fragments in both production methods (Table EV2).

## Mechanism of empty capsid reduction

We reasoned that AAVPure[Mfg] reduced empty capsid formation by eliminating the possibility of functional Rep and Cap expression in the absence of pCis, such as when certain HEK293 cells are co-transfected with only pHelper and pTrans, but not pCis, in triple transfection (Fig. 1A). However, another possibility is that the minicircle Cis construct is a better vector genome source for efficient packaging. To rule out this possibility, we devised a Trans-Cis coupled triple transfection system consisting of three plasmids (Fig. 3A, middle panel): (1) the standard pHelper, (2) pTrans-STOP that contains the attP/attB-flanked SV40 pA (three copies) in replacement of attR2, and (3) pCis-Bxb1 that provides the ITR-flanked transgene cassette and a separate *Bxb1* cassette. This new system does not generate a minicircle Cis construct but mimics AAVPure[Mfg] in that, when pHelper and pTrans-STOP coexist in a HEK293 cell nucleus without pCis-Bxb1, no capsid proteins are expressed because the SV40 pA prematurely terminates *Cap* transcription, thereby preventing empty capsid formation. Coexistence of all three plasmids enables Bxb1-mediated excision of SV40 pA, restoring Rep and Cap expression for vector packaging. As expected, replacing pCis-Bxb1 with the standard pCis failed to produce rAAV (Fig. 3B), demonstrating the tight Rep/Cap expression regulation conferred by pTrans-STOP.

We found that the Trans-Cis coupled triple transfection and AAVPure[Mfg] 1.0 showed similar characteristics in rAAV production regarding genome titer, capsid titer, and full capsid ratio (Fig. 3B–D). We also assessed the dynamics of *Cap* expression at both protein and mRNA levels in HEK293 cells, and found that it peaked at 48 h post transfection in all three production schemes (Fig. 3E,F). However, the expression levels in Trans-Cis coupled triple transfection and AAVPure[Mfg] 1.0 were two- to threefold lower than those in triple transfection across all time points examined (Fig. 3E,F), in agreement with the reduced capsid titers (Fig. 3C).

To determine whether lower Cap expression levels contributed to the improvement in full capsid ratio, we designed an inducible Cap expression system using the cumate-gated gene switch (Mullick et al, 2006) (Fig. EV2A–C). We found that reducing Cap expression resulted in a concomitant decrease in both vector genome titer and capsid titer, which led to a moderate increase in the full capsid ratio (Fig. EV2D,E). A similar inverse relationship between rAAV genome titer and full capsid ratio was reported in a previous study (Lee et al, 2022). Therefore, simply reducing Cap expression did not fully account for the advantages observed with AAVPure[Mfg].

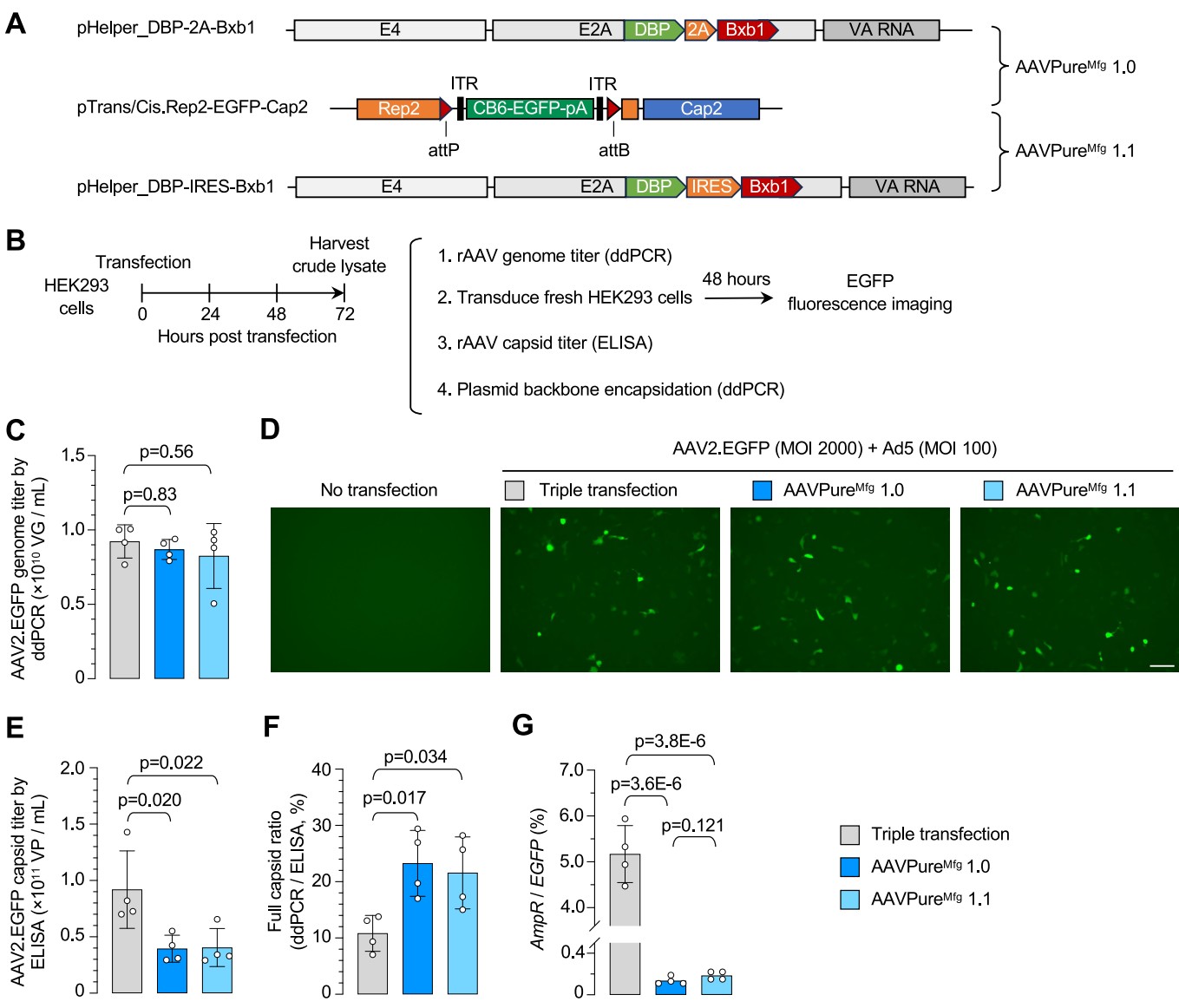

**Figure 2. AAVPure^Mfg improves AAV2 vector purity.**

(A) Schematics of plasmid components in AAVPure^Mfg 1.0 and AAVPure^Mfg 1.1. (B) Experimental procedure of rAAV production and vector characterization. (C) Packaging yield of AAV2.EGFP produced by either triple transfection (gray), AAVPure^Mfg 1.0 (dark blue), or AAVPure^Mfg 1.1 (light blue). Small-scale rAAV production was conducted in a 12-well plate with plasmid usage as described in Table EV1. Crude lysate was harvested 72 h post transfection followed by three successive freeze–thaw cycles. Cleared crude lysates after centrifugation were treated with DNase-I and proteinase K, followed by droplet digital PCR (ddPCR) to determine the genome titer. (D) Representative fluorescence images of HEK293 cells infected by rAAV-containing crude lysates. Equal multiplicity of infection (MOI) of rAAV produced by triple transfection, AAVPure^Mfg 1.0, or AAVPure^Mfg 1.1 were used to infect HEK293 cells in the presence of adenovirus 5 (Ad5). Images were taken 2 days post infection. Scale bar, 100 μm. (E) AAV.EGFP capsid titers in cleared lysates determined by ELISA assay. (F) Full capsid ratio determined by ddPCR genome titer normalized to ELISA capsid titer. (G) Plasmid backbone DNA levels in rAAV products. Duplex ddPCR was performed with one probe targeting *EGFP* transgene and the other for *AmpR* in cleared lysates treated with DNase-I and proteinase K. In (C, E–G), data are mean ± s.d. of biological replicates, $n = 4$ for each group. Statistical analysis was performed using one-way ANOVA followed by Dunnett's multiple comparisons test against the triple transfection group. Exact *P* value was indicated in the figure. Source data are available online for this figure.

In summary, these results indicate that asynchronous presence of pTrans and pCis is a major contributor to empty capsid formation in standard triple transfection. Coupling pTrans and pCis through either Trans-Cis coupled triple transfection or AAVPure^Mfg 1.0 prevents functional Rep/Cap expression in the absence of pCis, thereby reducing empty capsid formation and improving full capsid ratio without compromising rAAV genome titer.

## Developing AAVPure^Mfg 2.0 that utilizes standard pHelper

We sought to test whether AAVPure^Mfg is compatible with the standard pHelper when supplemented with a small amount of pHelper_DBP-2A-Bxb1 (pHelper-Bxb1 hereafter) (Fig. 4A). This design, designated as AAVPure^Mfg 2.0, aims to utilize the existing

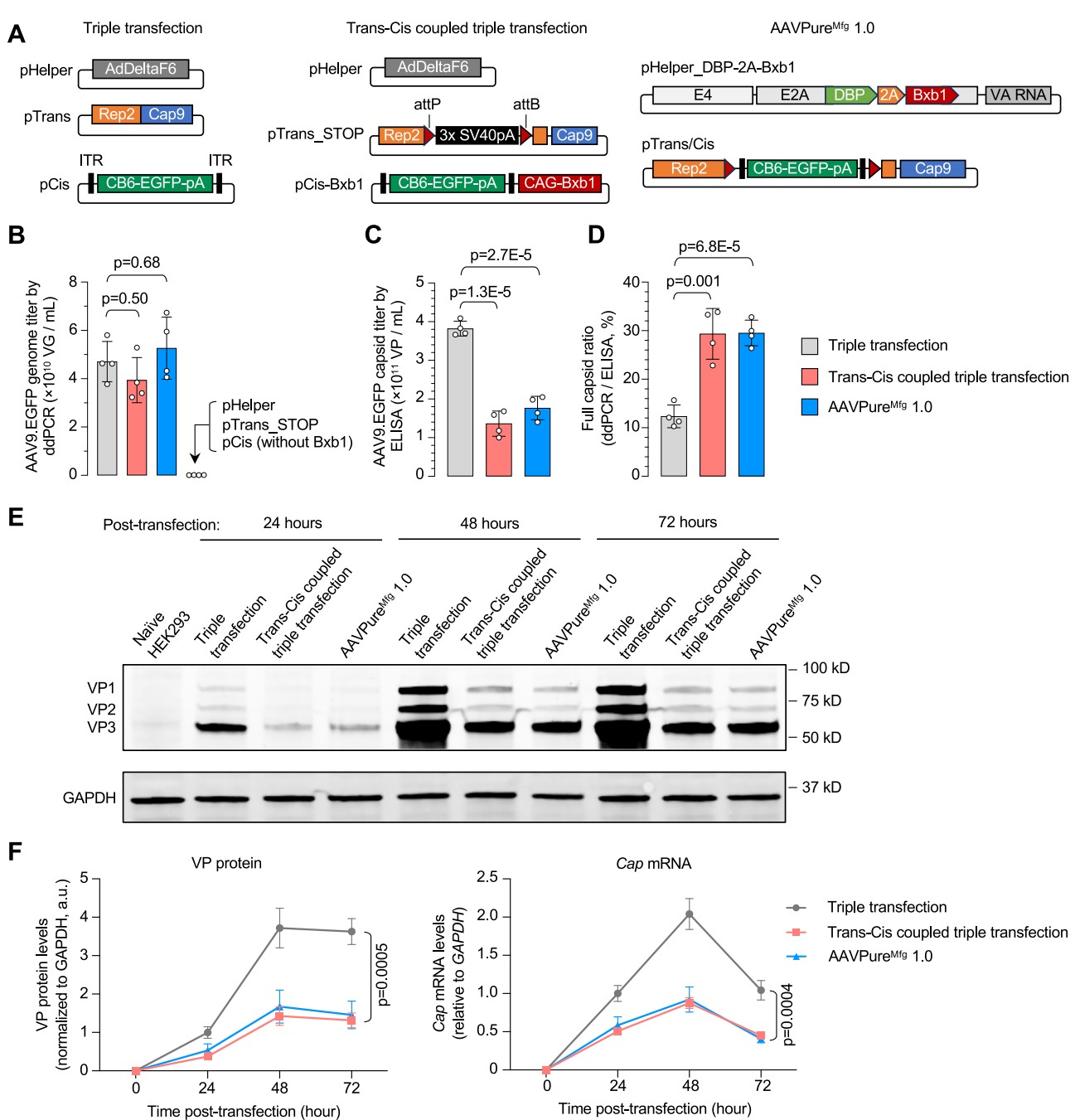

**Figure 3. Asynchronous presence of pTrans and pCis plasmids causes empty capsid formation in triple transfection.**

(A) Schematics of plasmid components in triple transfection (left panel), Trans-cis coupled triple transfection (middle panel) or AAVPure^Mfg 1.0 (right panel) used to produce AAV9.EGFP. (B) Comparison of AAV9.EGFP genome titer produced by different transfection methods. The rAAV production and titering procedures were the same as Fig. 2B. (C) AAV9.EGFP capsid titer in cleared lysate determined by ELISA assay. (D) Full capsid ratio determined by ddPCR genome titer normalized to ELISA capsid titer. (E) Western blotting of AAV9 VP proteins in HEK293 cells by different production methods and at indicated time points post transfection. (F) Dynamics of VP protein and *Cap* mRNA expression levels in HEK293 cells by different production methods and at indicated time points post transfection. The protein or mRNA abundance in triple transfection at 24 h post plasmid transfection is normalized to be 1. Statistical analysis was performed using two-way ANOVA followed by Dunnett's multiple comparisons test against the triple transfection group. In (B–D, F), data are mean ± s.d. of biological replicates, n = 4 for each group. Statistical analysis was performed using one-way ANOVA followed by Dunnett's multiple comparisons test against the triple transfection group. Exact P value was indicated in the figure. Source data are available online for this figure.

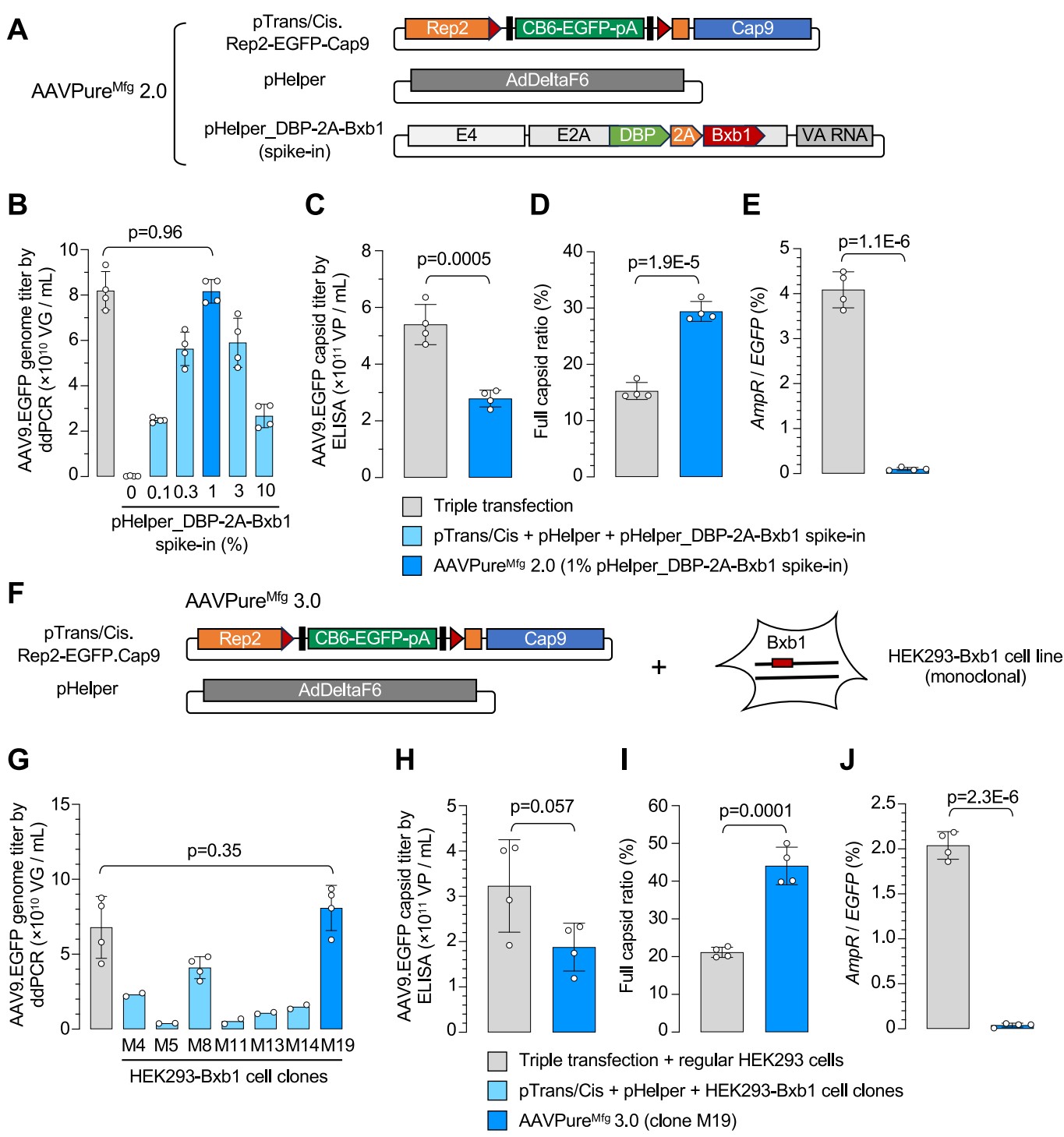

**Figure 4. Developing AAVPure^Mfg iterations that reduce plasmid manufacturing burden.**

(A) Schematics of plasmid components in AAVPure^Mfg 2.0. (B) AAV9.EGFP genome titer produced by triple transfection or AAVPure^Mfg 2.0 with different amounts of pHelper_DBP-2A-Bxb1 spike-in. The rAAV production and titering procedures were the same as Fig. 2B. (C–E) Comparison of AAV9 capsid titer (C), full capsid ratio (D), and plasmid backbone DNA levels (E) in cleared lysates between triple transfection and AAVPure^Mfg 2.0 with 1% pHelper_DBP-2A-Bxb1 spike-in. (F) Schematics of plasmid and cellular components in AAVPure^Mfg 3.0. (G) AAV9.EGFP genome titer produced by triple transfection or AAVPure^Mfg 3.0 with different monoclonal HEK293-Bxb1 cell lines. The rAAV production and titering procedures were the same as Fig. 2B. (H–J) Comparison of AAV9 capsid titer (H), full capsid ratio (I), and plasmid backbone DNA levels (J) in cleared lysates between triple transfection and AAVPure^Mfg 3.0 with the monoclonal HEK293-Bxb1 cell line M19. The detailed plasmid usage is described in Table EV1. In (B–E, G–J), data are mean ± s.d. of biological replicates, $n = 4$ for each group except in Fig. 4G group M4, M5, M11, M13, M14, $n = 2$. Statistical analysis was performed using unpaired t test (C–E, H–J) or one-way ANOVA followed by Dunnett's multiple comparisons test against the triple transfection group (B, G). The exact P value was indicated in the figure. Source data are available online for this figure.

pHelper in various established triple or dual transfection systems (Liu et al, 2024; Schnodt et al, 2016; van Lieshout et al, 2023; Xiao et al, 1998), and lower the manufacturing burden for pHelper-Bxb1.

We first screened for the optimal amount of pHelper-Bxb1 spike-in for producing AAV9.EGFP, and found that 1% (relative to the mass of standard pHelper) yielded the highest vector genome titer, which was comparable to that by triple transfection (Fig. 4B; Table EV1). Similar to AAVPure^Mfg 1.0, AAVPure^Mfg 2.0 improved the full capsid ratio by twofold (Fig. 4C,D), and drastically reduced plasmid backbone contamination by 40-fold compared with triple transfection (Fig. 4E). Similar results were observed when producing AAV vectors with various serotypes (Fig. EV3A,B), transgenes (Fig. EV3C,D) and ITR configurations (Fig. EV3E,F), indicating the broad applicability of AAVPure^Mfg 2.0 for producing high-purity AAV vectors. Interestingly, we observed that packaging different transgenes or genome configurations required distinct optimal conditions for plasmid input (Fig. EV3), suggesting that a systematic optimization step may be necessary to maximize rAAV production with AAVPure^Mfg 2.0. This could involve using a Design of Experiments (DoE) approach. Notably, we found similar levels of DpnI-resistant rAAV vector DNA in HEK293 cells undergoing triple transfection and AAVPure^Mfg 2.0, suggesting that the vector genome rescue and replication efficiencies between conventional pCis and the minicircle Cis construct (mcCis) were comparable (Hewitt et al, 2009) (Appendix Fig. S5).

Next, we investigated whether other Bxb1-expressing plasmids of smaller sizes could replace the large pHelper-Bxb1. To this end, we used pCAG-Bxb1 (Addgene, 51271) as the spike-in (Hermann et al, 2014) (Fig. EV4A; Table EV1). Interestingly, we found that as little as 0.1% pCAG-Bxb1 spike-in led to efficient rAAV production (Fig. EV4B), as compared with pHelper-Bxb1 that performed the best when supplemented at 1% (Fig. 4B). This was likely due to its smaller size that facilitated more efficient transfection, or the strong CAG promoter driving higher Bxb1 expression. Nevertheless, we designated using 1% pCAG-Bxb1 spike-in as AAVPure^Mfg 2.1, because it also afforded efficient rAAV production (Fig. EV4B). Consistent with previous AAVPure^Mfg iterations, rAAV produced by AAVPure^Mfg 2.1 showed a twofold increase in full capsid ratio (Fig. EV4C,D) and a 45-fold decrease in prokaryotic DNA contamination (Fig. EV4E).

### A stable HEK293-Bxb1 cell line for AAVPure^Mfg 3.0

The compatibility with various Bxb1-expressing plasmids prompted us to develop AAVPure^Mfg 3.0, which is centered around a stable monoclonal HEK293-Bxb1 cell line (Fig. 4F). This approach obviates Bxb1 plasmid usage, thus further decreasing rAAV manufacturing costs. Using a CRISPR-mediated homology-directed repair (HDR) strategy, we inserted *Bxb1* into the *AAVS1* safe harbor locus known to enable robust stable transgene expression (Shin et al, 2020). The HDR template includes an artificial splicing acceptor (SA) and P2A/T2A ribosomal skipping elements, so that both *Bxb1* and the puromycin resistance gene (*PuroR*) are under control of the endogenous *AAVS1* promoter, thereby avoiding the introduction of an exogenous promoter that is often subjected to silencing (Cabrera et al, 2022) (Fig. EV5A,B). We obtained 48 puromycin-resistant cell clones, 10 of which showed growth rates comparable to that of unmodified HEK293 cells; further analysis identified seven clones that exhibited robust Bxb1 recombinase activity (Fig. EV5C–E).

Next, the seven candidate HEK293-Bxb1 cell clones were evaluated for producing AAV9.EGFP following the AAVPure^Mfg 3.0 scheme (Fig. 4F). We found that clone M19 generated the highest rAAV genome titer among all cell clones, which was comparable to that produced by standard triple transfection using unmodified HEK293 cells (Fig. 4G). AAVPure^Mfg 3.0 using M19 improved full capsid ratio by twofold, and decreased prokaryotic DNA encapsidation by 51-fold (Fig. 4H–J).

### AAVPure^Mfg with suspension HEK293 cells

Finally, we tested AAVPure^Mfg using suspension HEK293 cells, a platform commonly used in large-scale rAAV manufacturing, including at the cGMP level (Grieger et al, 2016; Pupo et al, 2022) (Fig. 5A). Consistent with the results obtained with adherent HEK293 cells, AAVPure^Mfg 2.0 (i.e., with 1% pHelper-Bxb1 spike-in) produced similar AAV9.EGFP titer as standard triple transfection in suspension HEK293 cells (Fig. 5B). In contrast, the capsid titer was halved, resulting in a twofold increase in the full capsid ratio from 20.7% in triple transfection to 41.1% in AAVPure^Mfg 2.0 (Fig. 5C,D).

For rAAV generated by both methods, we purified the vector DNA to evaluate their size distribution by denaturing alkaline gel electrophoresis. Both vectors contained a 2.3 kb DNA species of expected vector genome size; however, a larger band corresponding to the double-sized genome (4.6 kb) was detectable only in the vector produced by triple transfection (Fig. 5E; Appendix Fig. S6), likely due to incomplete ITR resolution as previously reported for other rAAV preparations (Barnes et al, 2021; Tran et al, 2022). These data indicate that AAVPure^Mfg 2.0 produces rAAV with improved vector genome homogeneity. Various DNA impurities in purified rAAV were quantified by targeted droplet digital PCR (ddPCR). As demonstrated using adherent HEK293 cells, plasmid backbone DNA in the AAVPure^Mfg 2.0 vector showed a 45-fold reduction as compared with the triple transfection vector (Fig. 5F). The AAVPure^Mfg 2.0 vector contained lower or equal amount of other known DNA impurities, including the adenoviral helper genes *E2A* and *E4*, AAV *Rep* and *Cap* genes, and host cell DNA (Fig. 5G–I). Together, these data demonstrate that AAVPure^Mfg with suspension HEK293 cells effectively produces rAAV with an improved full capsid ratio and enhanced vector DNA purity.

## Discussion

Here, we developed AAVPure^Mfg, an improved rAAV manufacturing platform that addresses two critical limitations inherent to triple transfection: high levels of plasmid backbone contaminants and empty capsid formation. AAVPure^Mfg builds on a series of engineering steps that synergize Bxb1 recombinase, minicircle DNA, and the structural flexibility of the Rep C-terminus domain. To our knowledge, this is the first demonstration that a minicircle cis construct can be generated from a parental plasmid during rAAV production phase, and that this strategy markedly reduces plasmid backbone encapsidation to around 0.1%, an unprecedentedly low level in plasmid-based rAAV manufacturing. As inappropriately packaged bacterial DNA sequences were associated with inflammation and toxicities observed in some pre-clinical (Keiser et al, 2021) and clinical (Pena, 2018) settings, the

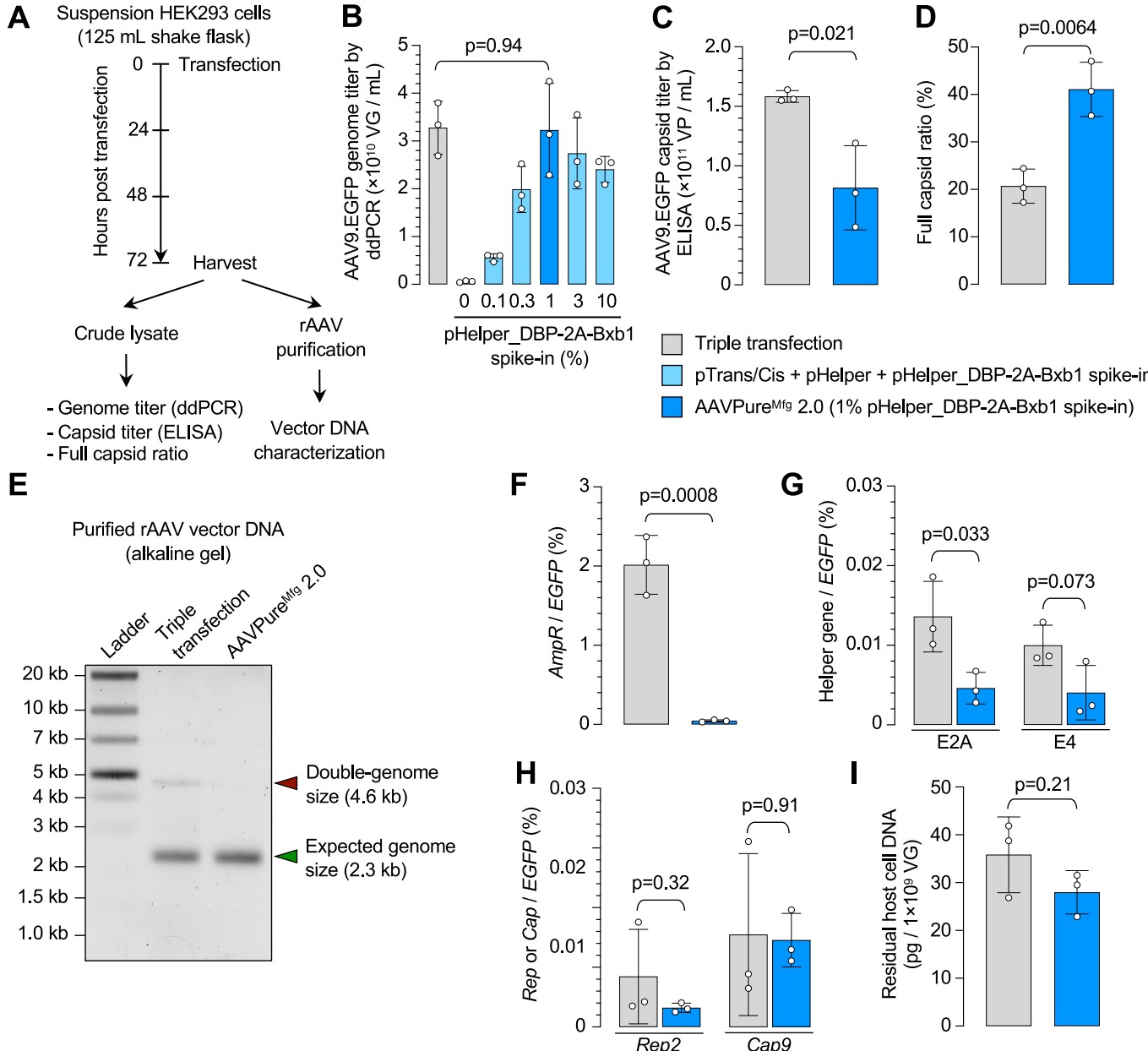

**Figure 5. Application of AAVPure^Mfg 2.0 to suspension HEK293 cells.**

(A) Experimental procedure of AAV9.EGFP production and vector characterization. (B) AAV9.EGFP genome titer produced by triple transfection or AAVPure^Mfg 2.0 with different amounts of pHelper_DBP-2A-Bxb1 spike-in. (C, D) Comparison of AAV9 capsid titer (C) and full capsid ratio (D) in cleared lysates between triple transfection and AAVPure^Mfg 2.0 with 1% pHelper_DBP-2A-Bxb1 spike-in. (E) Denaturing alkaline gel image showing the size of vector DNA purified from AAV9.EGFP in (A). Green arrowhead indicates the full-length vector genome size; red arrowhead indicates the double-genome size. (F–I) Vector DNA impurities of plasmid backbone (F), adenovirus helper genes (G), Rep and Cap genes (H), and host HEK293 cell genomic DNA (I) between triple transfection and AAVPure^Mfg 2.0 with 1% pHelper_DBP-2A-Bxb1 spike-in. In (B–D, F–I), data are mean ± s.d. of biological replicates, n = 3 for each group. Statistical analysis was performed using unpaired t test (C, D, F–I) or one-way ANOVA followed by Dunnett's multiple comparisons test against the triple transfection group (B). The exact P value was indicated in the figure. Source data are available online for this figure.

AAVPure^Mfg system has the potential to make AAV therapies better tolerated. In addition, our comparative studies suggest that a primary cause of empty capsid formation in triple transfection is the asynchronous presence of pTrans and pCis due to stochastic uptake of plasmids, and that their genetic coupling effectively mitigates empty capsid formation. Consequently, AAVPure^Mfg

boosts full capsid ratios up to threefold at harvest. These results reveal that, besides the temporal desynchronization between capsid synthesis and vector genome replication (Nguyen et al, 2021), spatial desynchronization between pTrans and pCis plays an important role in empty capsid formation in triple transfection. An improved full capsid ratio in crude cell lysate is expected to greatly

facilitate downstream purification (Joshi et al, 2021; Roach et al, 2024), thereby reducing manufacturing costs and enhancing vector potency. Finally, we demonstrate that AAVPure[Mfg] is broadly applicable across different serotypes, transgenes, ITR configurations, and cell culture systems.

Recently, van Lieshout et al developed a dual-plasmid system, pOXB, which combines pTrans and pCis into a single plasmid by inserting an ITR-flanked transgene cassette next to *Rep/Cap* in pTrans (van Lieshout et al, 2023). While this approach improves the full capsid ratio presumably due to the physical linkage of *Rep/Cap* and cis construct, it appears to do so to a lesser extent than AAVPure[Mfg]. Furthermore, the pOXB design still suffers from potential prokaryotic backbone encapsidation and risks generating rcAAV due to the close proximity between functional *Rep/Cap* genes and ITRs. In contrast, the pTrans/Cis used in AAVPure[Mfg] features an ITR-flanked transgene embedded within the *Rep/Cap* genes and abolishing their expression and function at the default stage, thereby reducing the likelihood of generating rcAAV, as supported by PacBio sequencing.

Besides the initial AAVPure[Mfg] design, AAVPure[Mfg] 2.0 utilizes the same pHelper as for triple transfection and a small amount of modified pHelper expressing Bxb1. As pHelper is universally used in various plasmid-based AAV manufacturing systems, this iteration is expected to greatly reduce the manufacturing burden for pHelper-Bxb1. Additionally, various Bxb1-expressing plasmids are compatible with AAVPure[Mfg], which further facilitates its implementation. However, we note that it is possible for a small Bxb1 expression cassette, such as the CAG-Bxb1 (3.5 kb) used in AAVPure[Mfg] 2.1, to be packaged into rAAV and expressed in recipients. From this perspective, the pHelper-Bxb1 used in AAVPure[Mfg] 2.0 is designed to be much safer, because the entire Bxb1 cassette is around 7 kb, well beyond the AAV packaging capacity. This concern is further mitigated in AAVPure[Mfg] 3.0, where *Bxb1* is stably integrated into the *AAVS1* safe harbor locus in HEK293 cells and driven by the endogenous promoter. Indeed, the Bxb1 and PuroR contaminants were below the detection limit of ddPCR, further supporting the rAAV purity achieved by AAVPure[Mfg] platforms (Appendix Fig. S7). Thus, we recommend AAVPure[Mfg] 2.1 for pilot testing, and AAVPure[Mfg] 2.0 or 3.0 for process development and scale-up rAAV manufacturing.

Several limitations exist in this work. First, the proposed mechanism of AAVPure[Mfg] is unlikely to explain the phenomenon that, in triple transfection, full capsid ratio at harvest may vary across different therapeutic genes, even when packaged into the same AAV capsid (Steininger et al, 2025). Such differences may be attributable to the transgene length, DNA sequence, or the impact of the transgene product on the cellular environment. Second, while some proof-of-concept optimizations have been conducted, integration of a systematic DoE approach may further maximize yield. Such a methodology has been shown to significantly improve the performance of plasmid-based rAAV manufacturing systems (Park et al, 2024; Zhao et al, 2020). Third, we performed ddPCR and ELISA to determine full capsid ratios in crude lysates across various tested conditions. Although this method is widely used for this purpose (Ohba et al, 2023; Steininger et al, 2025; Su et al, 2022), it may underestimate intermediate capsids that are partially filled with fragmented vector genomes, or overfilled capsids containing extended vector genomes. In this regard, analytical

ultracentrifugation (AUC) is considered the industry standard for comprehensively and quantitatively profiling the distribution of various capsid species. Future studies may involve scaling up production with bioreactors, combined with industry-standard downstream purification and AUC to quantify empty, full, intermediate, and overfilled capsids.

An unexpected finding from this study is the improved vector genome homogeneity in AAVPure[Mfg], evidenced by the reduction of double-sized vector genome commonly observed in rAAV preparations (Barnes et al, 2021; Tran et al, 2022) (Fig. 5E; Appendix Fig. S6). While the precise mechanism remains to be studied, one possibility is that genetic coupling of pTrans and pCis in AAVPure[Mfg] enhances the accessibility of the vector genome to Rep proteins, increasing the likelihood of ITR resolution. Another interesting finding is that the *Rep* coding region between P40 and intron is amenable to in-frame peptide insertion. The plasticity of this Rep region in the context of rAAV production is also described in a recent *Rep* saturation mutagenesis screen (Jain et al, 2024). Future work may exploit this unique region to engineer Rep protein variants as valuable tools for AAV biology studies.

In summary, AAVPure[Mfg] produces high-purity AAV vectors with markedly improved full capsid ratios, enhanced vector genome homogeneity, and dramatically reduced prokaryotic DNA contaminants. Coupled with reduced plasmid demand, AAVPure[Mfg] offers a valuable platform for manufacturing rAAV and has the potential to make high-quality AAV-based genetic medicines more accessible to patients.

# Methods

**Reagents and tools table**

| Reagent/resource | Reference or source | Identifier or catalog number |
|---|---|---|
| **Experimental models** | | |
| Adherent HEK293 Cells | ATCC | CRL-1573 |
| Suspension Expi293F cells | Thermo Fisher Scientific | A14528 |
| Monoclonal HEK293-Bxb1 cells | This study | N/A |
| **Recombinant DNA** | | |
| pCAG-Bxb1 | Addgene | 51271 |
| pHelper | Addgene | 112867 |
| pRep2/CapX | Addgene | 112865 |
| pAAV.CAG.LSL.EGFP | Addgene | 100047 |
| pAAV2/2-attR1 (pHL280b) | This study | Table EV3 |
| pAAV2/2-attR2 (pHL281b) | This study | Table EV3 |
| pAAV2/2-attR3 (pHL282b) | This study | Table EV3 |
| pHelper_DBP-2A-Bxb1 (pHL284) | This study | Table EV3 |
| pHelper_DBP-IRES-Bxb1 (pHL285) | This study | Table EV3 |
| pTrans/Cis.Rep2-EGFP-Cap2 (pHL300) | This study | Table EV3 |
| pTrans-STOP (pHL327) | This study | Table EV3 |
| pAAVS1 LHA-SA-P2A-BxB1-T2A-PuroR-bGH pA-RHA (pHL380) | This study | Table EV3 |

| Reagent/resource | Reference or source | Identifier or catalog number |
| --- | --- | --- |
| pTrans/Cis.Rep2-EGFP-Cap9 (pHL319) | This study | N/A |
| pTrans/Cis.Rep2-EGFP-Cap8 (pHL367) | This study | N/A |
| pTrans/Cis.Rep2-N.Cas9-Cap9 (pHL467) | This study | N/A |
| pTrans/scCis.Rep2-EGFP-Cap9 (pHL319c) | This study | N/A |
| **Antibodies** | | |
| Mouse anti-Rep | Origen Technologies | AM09104PU-N |
| Mouse anti-VP1/2/3 | PROGEN Biotechnik | 61058 |
| Rabbit anti-GAPDH | Abcam | ab9485 |
| **Oligonucleotides and other sequence-based reagents** | | |
| ddPCR probes | This study | Dataset EV1 |
| **Chemicals, enzymes, and other reagents** | | |
| Proteinase K | QIAGEN | 19133 |
| DMEM | Gibco | 11965-084 |
| Fetal Bovine Serum | Gibco | 26140-079 |
| Penicillin/Streptomycin | Thermo Fisher Scientific | 15140122 |
| Freestyle F17 media | Gibco | A13835 |
| Glutamax | Gibco | 35050061 |
| Lipofectamine 2000 | Thermo Scientific | 11668027 |
| PEI Max | PolySciences | 24765-2 |
| AAVpro Purification Kit | Takara Bio | 6675 |
| DNase-I | Roche Life Science | 4716728001 |
| Vericheck ddPCR HEK293 Residual DNA Quantification Kit | Bio-Rad | 12016814 |
| AAV2 Xpress ELISA Kit | PROGEN Biotechnik | PRAAV2XP |
| AAV8 Xpress ELISA Kit | PROGEN Biotechnik | PRAAV8XP |
| AAV9 Xpress ELISA Kit | PROGEN Biotechnik | PRAAV9XP |
| **Software** | | |
| GraphPad Prism | https://www.graphpad.com | |
| ImageJ | https://imagej.nih.gov/ij/index.html | |
| **Other** | | |
| PacBio | PacBio Sequel II | |

## Plasmid construction

Plasmids were constructed using Gibson Assembly. The plasmids pAAV2/2-attR1 (pHL280b), pAAV2/2-attR2 (pHL281b), and pAAV2/2-attR3 (pHL282b) were generated by digesting the packaging plasmid pAAV2/2 with HindIII and subsequently assembling the digestion products with corresponding gBlocks from Integrated DNA Technologies (IDT), using NEBuilder HiFi DNA Assembly Master Mix (NEB, E2621L). For the construction of

pHelper_DBP-2A-Bxb1 (pHL284) and pHelper_DBP-IRES-Bxb1 (pHL285), a NsiI digestion site was introduced following the stop codon of DBP in the pHelper plasmid (Addgene, 112867). The 2A-Bxb1 and IRES-Bxb1 DNA fragments were then assembled via Gibson Assembly, with the 2A and IRES fragments carried in IDT gBlocks. The *Bxb1* gene was amplified via PCR from pCAG-Bxb1 (Addgene, 51271) using Phusion High-Fidelity DNA Polymerase (NEB, M0530S). pTrans/Cis plasmids were generated by digestion of pCis plasmid pAAV.CB6-EGFP with PacI, and then assembling this ITR-flanked transgene DNA with pAAV2/2, pAAV2/8, or pAAV2/9 packaging plasmids by Gibson Assembly at the attR2 site. Similar strategies were used to generate pTrans_STOP (pAAV2/9-3XSV40 pA, pHL327) with the 0.8 kb 3XSV40 pA fragment obtained from HindIII and BamHI digestion of pAAV.CAG.LSL.EGFP (Addgene, 100047). The left homologous arm (LHA)-splicing acceptor (SA)-P2A, T2A-PuroR-bGH pA, and right homologous arm (RHA) fragments in the donor plasmid of pAAVS1 LHA-SA-P2A-BxB1-T2A-PuroR-bGH pA-RHA (pHL380) were ordered from IDT as gBlocks. Bxb1 DNA fragment was also generated through PCR from pCAG-Bxb1 (Addgene, 51271) by Phusion High-Fidelity DNA Polymerase (NEB, M0530S). These fragments were then assembled into a donor plasmid by Gibson Assembly. The detailed plasmid sequences can be found in Table EV3.

## Cell culture

Adherent HEK293 cells (CRL-1573) were obtained from ATCC and maintained in DMEM (Gibco, 11965-084) supplemented with 10% (v/v) Fetal Bovine Serum (FBS; Gibco, 26140-079) and 1% (v/v) penicillin/streptomycin (Thermo Fisher Scientific, 15140122) at 37 °C in a humidified atmosphere containing 5% $CO_2$. Suspension Expi293F cells, purchased from Thermo Fisher Scientific (A14528), were cultured in Freestyle F17 media (Gibco, A13835) supplemented with 10 mM Glutamax (Gibco, 35050061) at 37 °C with 5% $CO_2$, 80% humidity, and shaking at 120 rpm.

## Monoclonal HEK293-Bxb1 cell line generation

For the generation of the HEK293-Bxb1 cell line, adherent HEK293 cells were transfected using a mixture of a Cas9-expressing plasmid, an AAVS1-targeting sgRNA plasmid (from Dr. Alex Brown), and the donor plasmid pAAVS1 LHA-SA-P2A-BxB1-T2A-PuroR-bGH (pHL380) at a mass ratio of 1:1:3. Transfection was performed using Lipofectamine 2000 (Thermo Scientific, 11668027). Three days post transfection, the cells were split and selected with puromycin (1 µg/mL) for 7 days. Monoclonal colonies were isolated through serial dilution and confirmed via a Bxb1 function assay.

## AAV vector production using adherent HEK293 cells

Small-scale vector preparations were generated in 12-well plates. In triple transfection, HEK293 cells were transfected with three plasmids carrying the vector genome (pCis), Rep/Cap (pTrans, pRep2/CapX; Addgene #112865) and adenovirus helper genes (pHelper; Addgene #112867), respectively, at equal mass ratio of 1:1:1 using the calcium phosphate method (Promega, E1200), totaling 1.5 µg/well. In AAVPure[Mfg], the detailed plasmid sequence of pHelper-Bxb1 and pTrans/Cis plasmids can be found in

Table EV3. pCAG-Bxb1 was purchased from Addgene (Plasmid #51271). pHelper-Bxb1 and pTrans/Cis were transfected with equal moles as that of pHelper and pTrans in HEK293 cells as described in Table EV1. Transfected cells were maintained in DMEM supplemented with 10% FBS and 1% penicillin/streptomycin for 24 h, after which the culture medium was replaced with DMEM supplemented with 1% penicillin/streptomycin without FBS. 72 h post transfection, cells and culture media were harvested, and subjected to 3 successive freeze–thaw cycles. The crude lysates were centrifuged at $14,000\times g$/min for 15 min at 4 °C to remove cell debris. Cleared crude lysates were treated with DNase-I and proteinase K, and used in a droplet digital PCR (ddPCR) assay to determine titer.

Medium-scale AAV vectors were produced following a similar procedure in 15-cm dishes containing 30 mL culture media. At 72 h after transfection, AAV9.EGFP vectors were purified by the AAVpro Purification Kit (Takara Bio, 6675). Vector genome titer and plasmid backbone contaminants in purified rAAV were quantified using ddPCR. Purified vector DNA was used for PacBio sequencing.

## AAV vector production using suspension Expi293F cells

In total, 30 mL Expi293F cells (Thermo Scientific, A14528) cultured in a 125 mL shaker flask were transfected at a density of $1 \times 10^6$ cells/mL using PEI Max (PolySciences, 24765-2). For triple transfection, three plasmids were co-transfected: one carrying the vector genome (pCis, 10 μg), one carrying Rep/Cap (pTrans, pRep2/CapX, 10 μg), and one carrying adenovirus helper genes (pHelper, 10 μg). For AAVPure^Mfg 2.0, the three plasmids were pHelper (10 μg), pTrans/Cis (13 μg) and spike-in pHelper_DBP-2A-Bxb1 (ranging from 0.01 μg (0.01%) to 1ug (10%) as indicated). Transfected cells were maintained in Freestyle F17 media supplemented with 10 mM Glutamax. 72 h post transfection, a small aliquot of cells (1 mL) was used to prepare cleared lysate for determining genome and capsid titers. The remaining cells (29 mL) were purified using the AAVpro Purification Kit (Takara Bio, 6675) for vector DNA characterization.

## Quantification of vector DNA impurities by ddPCR

rAAV crude lysates or purified AAV vectors were treated with DNase-I (Roche Life Science, 4716728001) and proteinase K (QIAGEN, 19133). Duplexing Taqman ddPCR assays were performed with one reagent targeting *EGFP* transgene (Thermo Fisher Scientific, Mr00660654_cn), and the other targeting *AmpR*, adenoviral helper genes, or AAV *Rep/Cap* genes. The vector DNA impurity was calculated by normalizing its value to that of *EGFP*. Detailed ddPCR probe sequences can be found in Dataset EV1. Residual host cell DNA in purified AAV vector was quantified by Vericheck ddPCR HEK293 Residual DNA Quantification Kit (Bio-Rad, 12016814) following the manufacturer's instructions.

## Quantification of rAAV full capsid ratio

rAAV genome titer (vg/mL) in crude lysate was quantified by ddPCR using a Taqman reagent targeting the *EGFP* transgene (Thermo Fisher Scientific, Mr00660654_cn) post treatment with DNase-I (Roche Life Science, 4716728001) and proteinase K (QIAGEN, 19133). rAAV particle concentration (pt/mL) in crude lysate was quantified using the AAV2 Xpress ELISA Kit (PROGEN Biotechnik, PRAAV2XP), AAV8 Xpress ELISA Kit (PROGEN Biotechnik, PRAAV8XP), or AAV9 Xpress ELISA Kit (PROGEN Biotechnik, PRAAV9XP) following the manufacturer's manual. Full capsid ratio is calculated as genome titer divided by capsid titer.

## Western blotting

Cultured cells were pelleted by centrifugation at $1000\times g$/min for 5 min at 4 °C, and then lysed with M-PER (Thermo Fisher Scientific, 78501) with proteinase inhibitor (Roche, 4693159001). Protein concentration was determined using Pierce BCA Protein Assay Kit (Pierce, 23225). Normalized protein lysates were boiled for 10 min in reducing SDS sample buffer (Boston BioProducts, BP-111R). Primary antibodies: mouse anti-Rep (Origen Technologies, AM09104PU-N, 1:100), mouse anti-VP1/2/3 (PROGEN Biotechnik, 61058, 1:200), rabbit anti-GAPDH (Abcam, ab9485, 1:2000). Secondary antibodies: LI-COR IRDye 680RD goat anti-mouse IgG (H + L) (LI-COR Biosciences, 926-68070, 1:3000), LI-COR IRDye 800CW goat anti-rabbit IgG (H + L) (LI-COR Biosciences, 926-32211, 1:3000). Blot membranes were imaged by LI-COR scanner (Odyssey) and quantified by ImageJ Fiji.

## cDNA quantification by ddPCR

RNA from transfected HEK293 cells was extracted by TRIzol reagent (Life Technologies, 15596-018). Purified RNA was reverse-transcribed using the High-Capacity cDNA Reverse Transcription kit (Thermo Fisher Scientific, 43-688-13). *Cap9* cDNA was quantified by duplex ddPCR with one Taqman reagent targeting *Cap9* as described in Dataset EV1 and the other targeting human *GAPDH* (Thermo Scientific, 4448490).

## AAV vector DNA analysis by alkaline agarose gel electrophoresis

In total, 0.8% agarose gel was prepared by boiling agarose in ultra-pure water, followed by cooling to 55 °C and adding 0.1 volume of 10× alkaline gel electrophoresis buffer (500 mM NaOH and 10 mM EDTA). In all, 200 μl of purified rAAV was treated with DNase-I and proteinase K, and then purified by phenol: chloroform: isoamyl alcohol solution. Purified vector DNA was mixed with 6X alkaline gel loading buffer (Thermo Fisher Scientific, AAJ62157AB), and loaded to alkaline gel. Electrophoresis was performed at a voltage of 3 V/cm for approximately 3 h. Then the gel was soaked in neutralization solution (BioWorld, 10750014) for 1 h at room temperature. The neutralized gel was stained with SYBR Gold (1:10,000 dilution, Thermo Fisher Scientific, S-11494) in 1× TAE buffer for 15 min, and imaged using a Bio-Rad Gel Doc XR+ Imaging System.

## PacBio sequencing and bioinformatic analysis

PacBio sequencing and bioinformatic analysis were described previously (Liu et al, 2024; Tai et al, 2018). Briefly, rAAV genome was purified and subjected to singe molecule, real-time (SMRT) sequencing at the Deep Sequencing Core of University of Massachusetts Chan Medical School. Sequencing reads were

de-multiplexed, then mapped to relevant plasmid and host genome references using the Burrows–Wheeler aligner-maximal exact match tool.

## Graphics

Graphics were created with BioRender.com.

## Statistical analysis

Data were presented as mean ± standard deviation (SD) of biological replicates. Comparison among two groups was analyzed by unpaired $t$ test. Comparison among multiple groups was analyzed by one-way analysis of variance (ANOVA) followed by Dunnett's multiple comparisons test. GraphPad Prism 10 was used for statistical analysis and data plotting.

# Data availability

The datasets produced in this study are available in the following databases: [PacBio]: [NCBI SRA] [PRJNA1239179].

The source data of this paper are collected in the following database record: biostudies:S-SCDT-10_1038-S44321-025-00248-w.

# Peer review information

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

## Acknowledgements

We wish to thank Dr. Alexander Brown for the Cas9 and AAVS1-targeting sgRNA constructs, and Dr. Jun Xie for designing the *AmpR* Taqman reagent. We wish to thank Deep Sequencing Core of the University of Massachusetts Chan Medical School for the service. The Wang Lab is supported by grant from the National Institutes of Health (NIH) (P01HL158506). The Gao Lab is supported by grants from the NIH (R01NS076991-01, P01AI100263-01, P01HL131471-02, 35 R01AI121135, UG3HL147367-01, R01HL097088, and U19AI149646-01) and the Cystic Fibrosis Foundation. This work was funded by Pfizer Inc. under a Sponsored Research Agreement.

## Author contributions

**Hao Liu**: Conceptualization; Resources; Data curation; Formal analysis; Validation; Investigation; Visualization; Methodology; Writing—original draft; Project administration; Writing—review and editing. **Nan Liu**: Resources; Methodology. **Chen Zhou**: Resources; Methodology. **Ailing Du**: Resources; Methodology. **Mayank Kapadia**: Resources; Methodology. **Phillip WL Tai**: Data curation; Formal analysis; Methodology. **Erik Barton**: Formal analysis; Supervision; Funding acquisition; Project administration. **Guangping Gao**: Formal analysis; Supervision; Funding acquisition; Writing—original draft; Project administration; Writing—review and editing. **Dan Wang**: Formal analysis; Supervision; Funding acquisition; Investigation; Visualization; Writing—original draft; Project administration; Writing—review and editing.

Source data underlying figure panels in this paper may have individual authorship assigned. Where available, figure panel/source data authorship is listed in the following database record: biostudies:S-SCDT-10_1038-S44321-025-00248-w.

## Disclosure and competing interests statement

HL, GG, and DW are inventors of a patent application filed by the University of Massachusetts Chan Medical School concerning the work described in this

study. EB was an employee of Pfizer Inc. when the study was conducted. GG is a scientific co-founder of Voyager Therapeutics, Adrenas Therapeutics and Aspa Therapeutics, and holds equity in these companies.

# Expanded View Figures

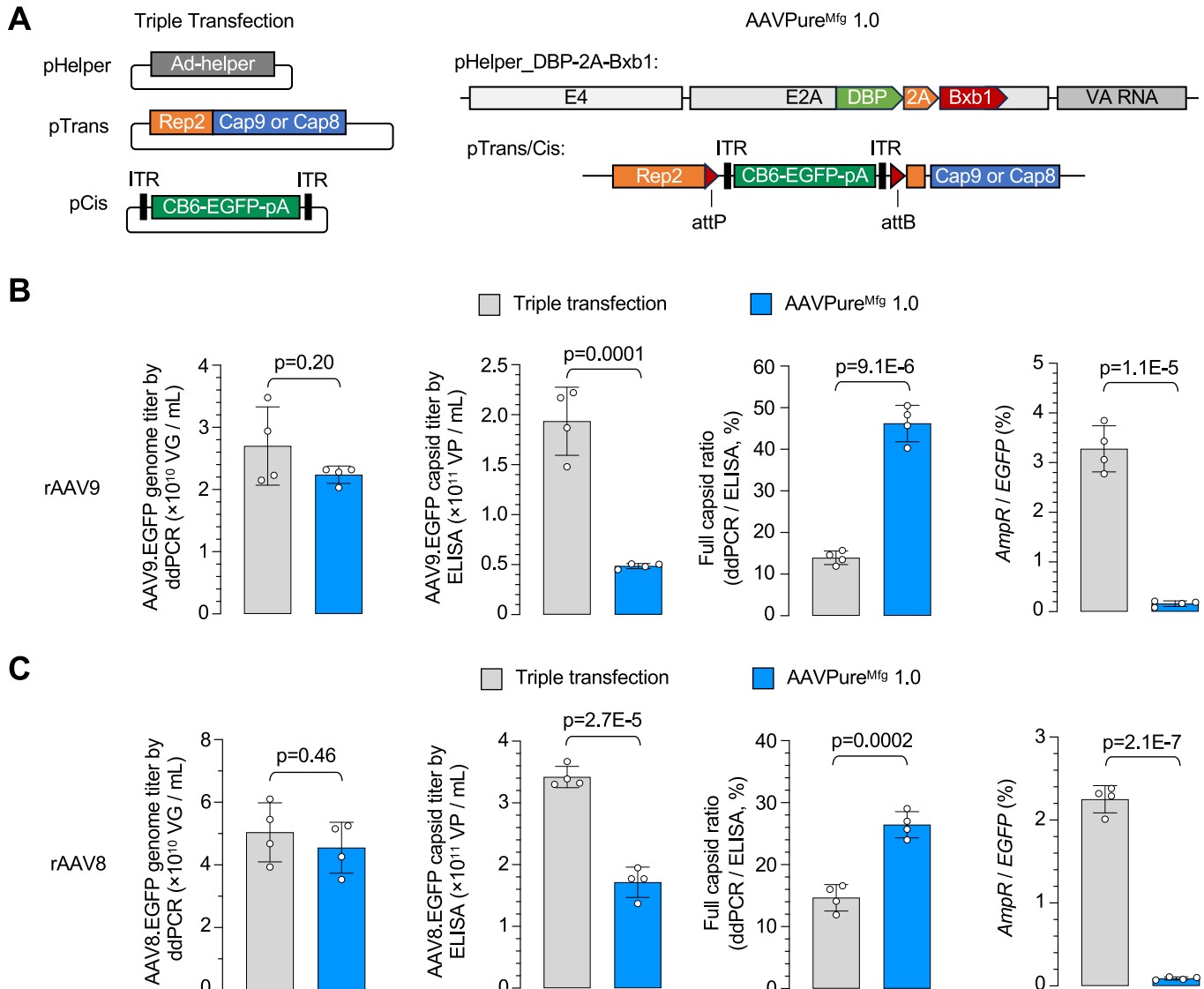

**Figure EV1. Application of AAVPure^Mfg 1.0 in different AAV serotypes.**

(A) Schematics of plasmid components in triple transfection (left panel) and AAVPure^Mfg 1.0 (right panel) used to produce AAV9.EGFP and AAV8.EGFP vectors.
(B) Comparison of AAV9.EGFP produced by either triple transfection or AAVPure^Mfg 1.0. (C) Comparison of AAV8.EGFP produced by either triple transfection or AAVPure^Mfg 1.0. The procedure of rAAV production and vector characterization in (B, C) were the same as shown in Fig. 2. In (B, C), data are mean ± s.d. of biological replicates, $n = 4$ for each group. Statistical analysis was performed using unpaired $t$ test. Exact $P$ value was indicated in the figure.

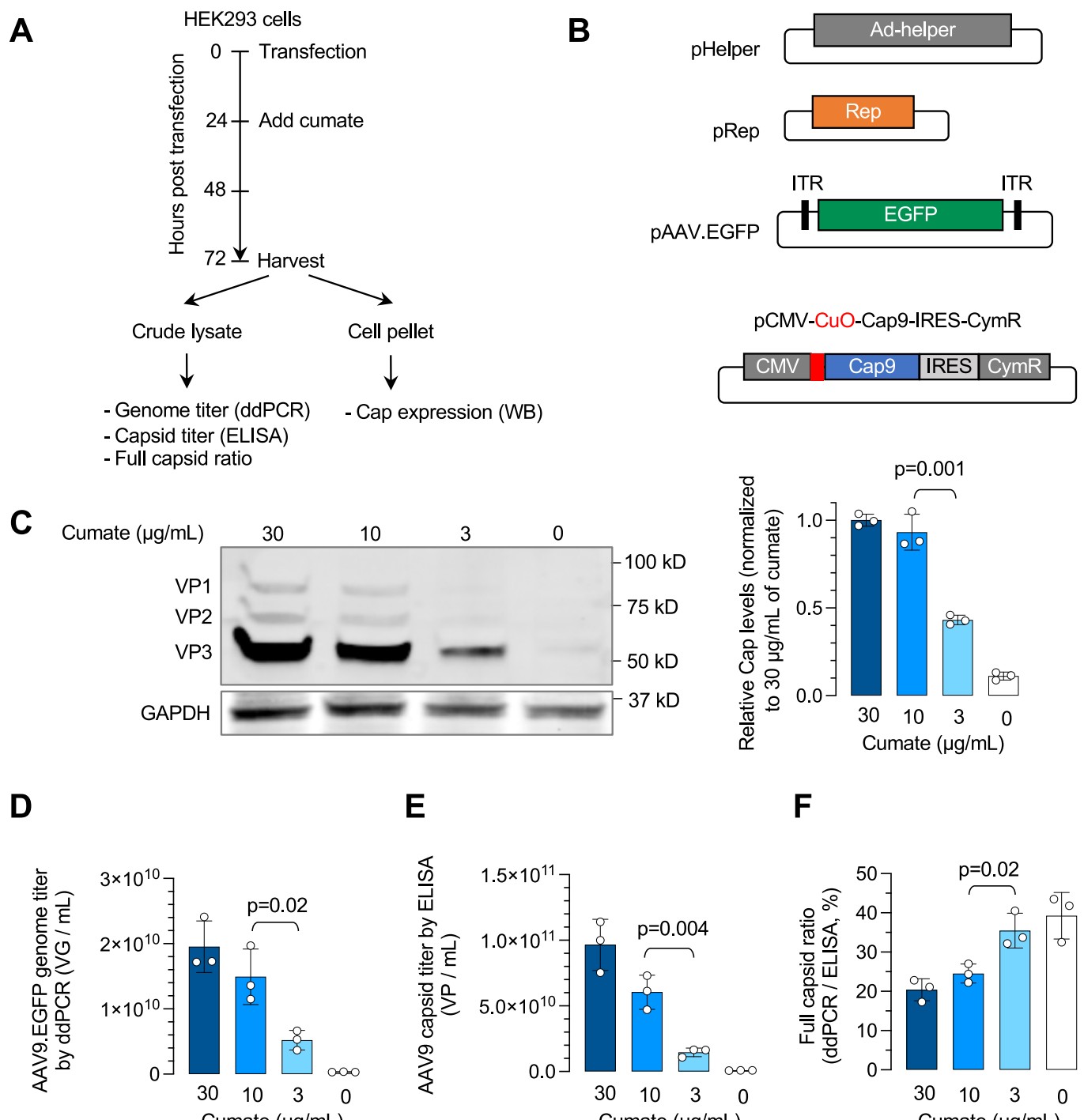

**Figure EV2. Lower VP expression increased the full capsid ratio with a marked decrease in vector genome titer.**

(A) Schematic showing the experimental procedure. (B) Construct illustrations of the plasmids used in quadruple transfection to produce AAV9.EGFP. (C) Representative western blotting images (left) and the quantification (right) of viral proteins (VP1, 2, 3) at indicated cumate concentrations. (D–F) Comparison of rAAV genome titer, capsid titer, and full capsid ratio in cleared lysates with different cumate concentrations. In (C–F), data are mean ± s.d. of biological replicates, $n = 3$ for each group. Statistical analysis was performed using unpaired $t$ test. Exact $P$ value was indicated in the figure.

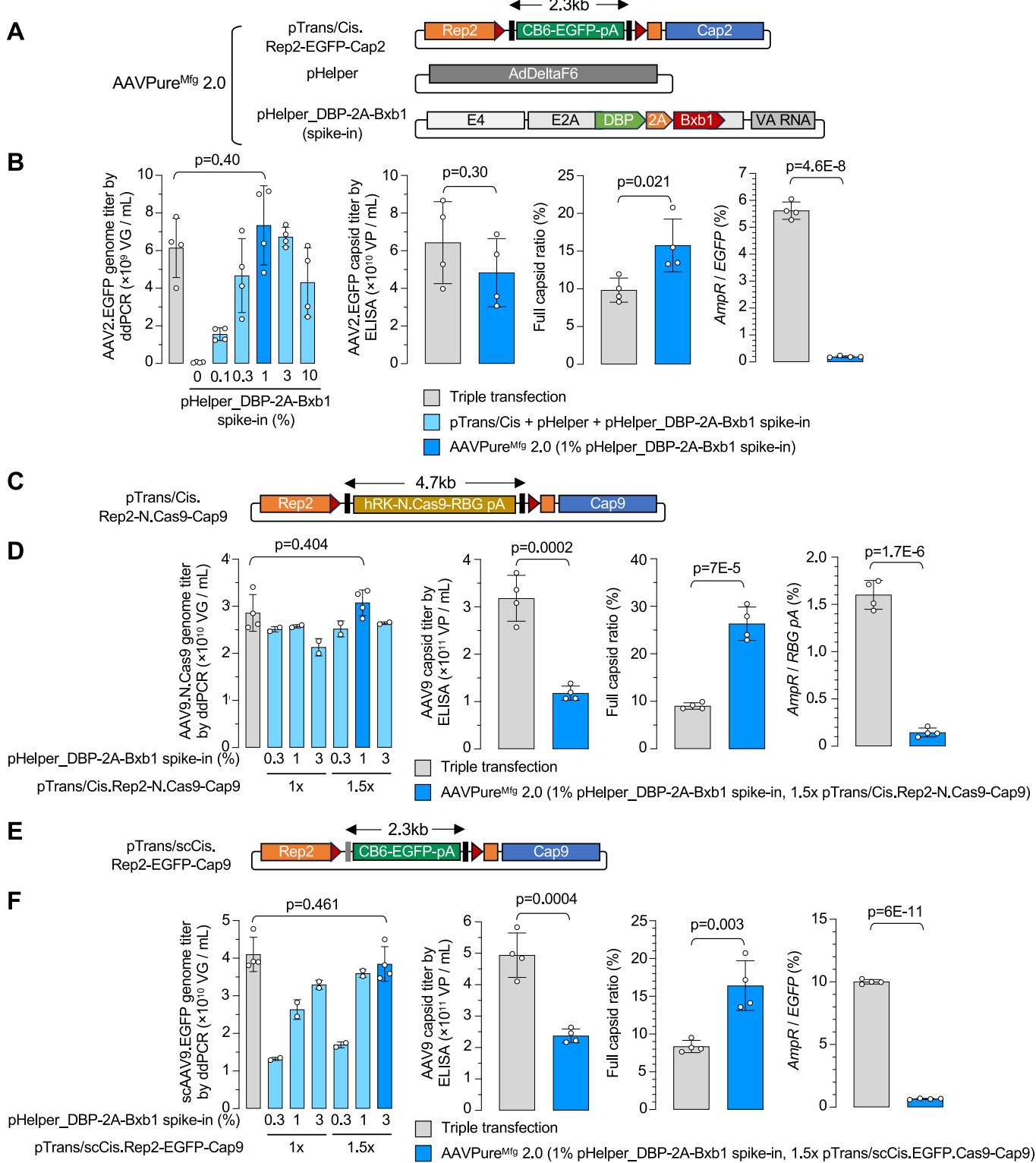

**Figure EV3. Application of AAVPure^Mfg 2.0 in producing AAV vectors with different serotypes, transgenes, and ITR configurations.**

(A, C, E) Schematics of plasmid components in AAVPure^Mfg 2.0 in producing ssAAV2.EGFP, ssAAV9.N.Cas9, and scAAV9.EGFP. In (E), the gray bar indicates mutant ITR that results in self-complementary (sc) vector genome. (B, D, F) Comparison of rAAV genome titer, capsid titer, full capsid ratio, and plasmid backbone DNA levels in cleared lysates between triple transfection and AAVPure^Mfg 2.0. In (B, D, F), data are mean ± s.d. of biological replicates, $n = 4$ for each group. Statistical analysis was performed using unpaired $t$ test. Exact $P$ value was indicated in the figure.

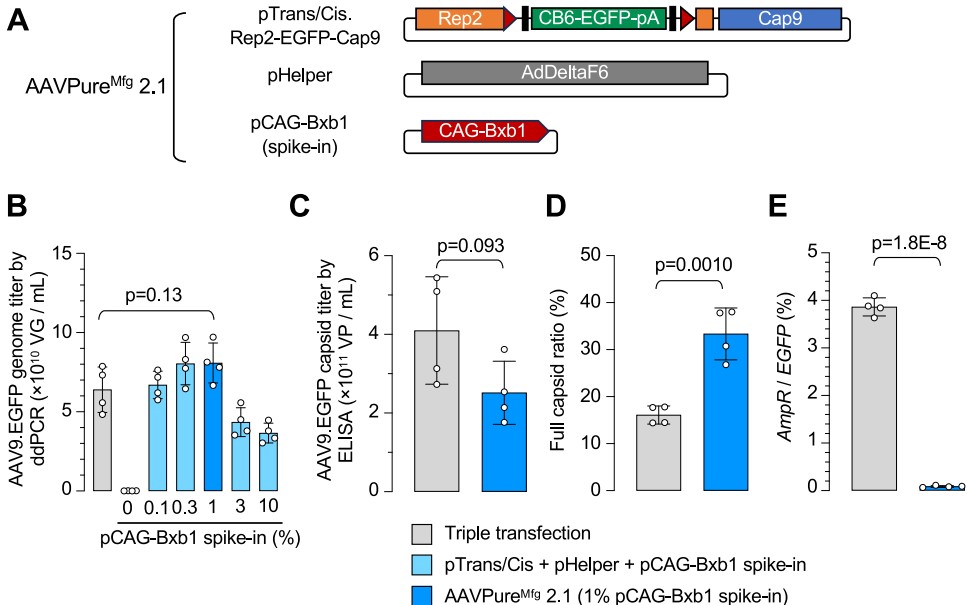

**Figure EV4. Developing AAVPure^Mfg 2.1 with pCAG-Bxb1 spike-in.**

(A) Schematics of plasmid components in AAVPure^Mfg 2.1. (B) AAV9.EGFP genome titer produced by triple transfection or AAVPure^Mfg 2.1 with different spike-in amount of pCAG-Bxb1. The rAAV production and titering procedures were the same as Fig. 2B. (C–E) Comparison of AAV9 capsid titer (C), full capsid ratio (D), and plasmid backbone DNA levels (E) in cleared lysates between triple transfection and AAVPure^Mfg 2.1 with 1% pCAG-Bxb1 spike-in. The detailed plasmid usage is described in Table EV1. In (B–E), data are mean ± s.d. of biological replicates, $n = 4$ for each group. Statistical analysis was performed using unpaired $t$ test (C–E) or one-way ANOVA followed by Dunnett's multiple comparisons test against the triple transfection group (B). Exact $P$ value was indicated in the figure.

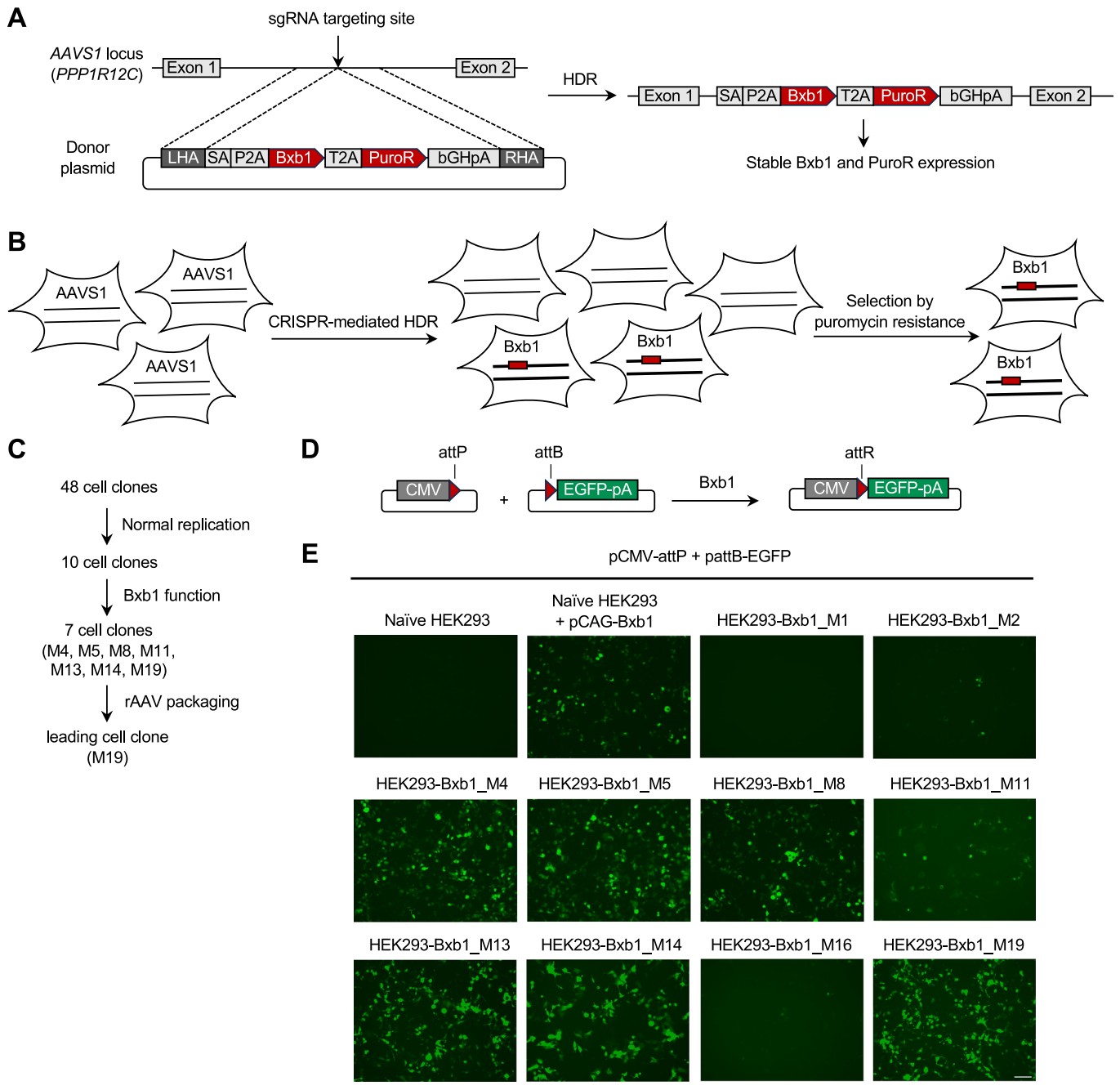

**Figure EV5. Generating a monoclonal HEK293-Bxb1 cell line.**

(A) Schematic diagram illustrating *Bxb1* knock-in into the *AAVS1* site in HEK293 cells. (B) Schematic diagram showing the workflow to engineer and select HEK293-Bxb1 cells. (C) Procedure of HEK293-Bxb1 monoclonal cell line generation and screening. (D) Schematics of a reporter assay to determine Bxb1 recombination activity. (E) Representative fluorescence images showing naive HEK293 cells or HEK293-Bxb1 monoclonal cell lines that were transfected with the reporter plasmids, pCMV-attP and pattB-EGFP. Images were taken 1 day post transfection. Scale bar, 100 µm.

 