## [Peer Review File · EMBO Molecular Medicine]

High-purity AAV vector production utilizing recombined minicircle formation and genetic coupling

Hao Liu, Nan Liu, Chen Zhou, Ailing Du, Mayank Kapadia, Phillip Tai, Eric Barton, Guangping Gao, and Dan Wang

Corresponding author(s): Dan Wang (Dan.Wang@umassmed.edu), Guangping Gao (Guangping.Gao@umassmed.edu)

Review Timeline:

Submission Date:	5th Dec 24
Editorial Decision:	9th Dec 24
Appeal:	10th Dec 24
Editorial Decision:	24th Dec 24
Revision Received:	2nd Apr 25
Editorial Decision:	12th Apr 25
Revision Received:	23rd Apr 25
Accepted:	25th Apr 25

Editor: Zeljko Durdevic

Transaction Report:

9th Dec 2024

Decision on your manuscript EMM-2024-21046

Dear Dr. Wang,

Thank you for the submission of your manuscript to EMBO Molecular Medicine. I have now carefully read your article and discussed it with the other members of our editorial team. I regret to say that we find the article not well suited for publication in EMBO Molecular Medicine and therefore have decided not to proceed with peer review.

The manuscript reports development of a method for high-purity AAV vector production. We do recognize the potential interest of the study for the immediate field; however, previous studies showing that minicircle technology improves purity of AAV preparations detract from the sort of the conceptual advance we would expect in an EMBO Molecular Medicine article. In our opinion, the findings are rather specialized for the broad audience of EMBO Molecular Medicine and would therefore be better suited for a more specialized venue. Therefore, I am afraid that we cannot offer further consideration to your article.

I would like to point out that the journal receives many more manuscripts than it can hope to publish and we have to make a correspondingly stringent selection of manuscripts we think likely both to appeal to our general readership and to fare well in the review process. This is not the case on this occasion and, in the interest of your time, I am providing you with an early decision on your manuscript that will allow you to submit it elsewhere without delay.

I am sorry that I can't be more positive on this occasion. Please rest assured that this is not a judgment of the quality or interest of your work, but a decision based on appropriateness for EMBO Molecular Medicine.

Yours sincerely,

Zeljko Durdevic

As a service to authors, EMBO provides authors with the possibility to transfer a manuscript that one journal cannot offer to publish to another EMBO publication. The full manuscript and if applicable, reviewers reports are automatically sent to the receiving journal to allow for fast handling and a prompt decision on your manuscript. For more details of this service, and to transfer your manuscript to another EMBO title please click on Link Not Available

Dear Dr. Durdevic,

Thank you for taking the time to evaluate our manuscript and for discussing it with your editorial team. While we are naturally disappointed with the decision, we would like to take this opportunity to clarify the novelty and significance of our study for your consideration.

The use of minicircles for AAV vector production was first reported nearly 9 years ago (*Molecular Therapy-Nucleic Acids*, 2016, 5, e355). However, this approach has not been widely adopted, primarily due to the complexities involved in minicircle preparation. This challenge is well documented, as noted by Dr. Jude Samulski et al., who stated: "However, both the fermentation process and the purification methods used for minicircle production are quite complex to set up" (*Molecular Therapy*, 2022, 30:12). In our study, a key innovation is the generation of minicircle DNA *in cellulo* using a standard plasmid as the starting material, effectively overcoming the major barrier to minicircle use in AAV vector manufacturing.

Furthermore, our comparative studies provide new insights into the origin of empty capsid formation. We have developed an AAV vector production method designed to eliminate this source, thereby improving the full capsid ratio—a long-sought goal in rAAV production.

Taken together, we believe these two significant advances—generation of minicircle DNA *in cellulo* and improvement of the full capsid ratio—represent valuable contributions to the field of AAV vector manufacturing. We are confident that these findings could help propel progress in gene therapy.

We respectfully ask that you reconsider our manuscript in light of these clarifications.

Sincerely,

Dan

--

Dan Wang, Ph.D.

Assistant Professor

Horae Gene Therapy Center | RNA Therapeutics Institute

University of Massachusetts Chan Medical School

N8-1027, 366 Plantation Street

Worcester, MA 01605

Phone: (774)-455-4574

E-mail: dan.wang@umassmed.edu

Website: GTC | RTI | Lab

24th Dec 2024

Dear Dr. Wang,

Thank you for the submission of your manuscript to EMBO Molecular Medicine. We have now received feedback from the three reviewers who agreed to evaluate your manuscript. All three referees recognize interest of the study but also raise important concerns that should be addressed in a major revision. If you would like to discuss further the points raised by the referees, I am available to do so via email or video. Let me know if you are interested in this option.

We would welcome the submission of a revised version within three months for further consideration. Please let us know if you require longer to complete the revision.

I look forward to receiving your revised manuscript.

Yours sincerely,

Zeljko Durdevic

We require:

- 1) A .docx formatted version of the manuscript text (including legends for main figures, EV figures and tables). Please make sure that the changes are highlighted to be clearly visible.
- 2) Individual production quality figure files as .eps, .tif, .jpg (one file per figure). For guidance, download the 'Figure Guide PDF': (<https://www.embopress.org/page/journal/17574684/authorguide#figureformat>).
- 3) A .docx formatted letter INCLUDING the reviewers' reports and your detailed point-by-point responses to their comments. As part of the EMBO Press transparent editorial process, the point-by-point response is part of the Review Process File (RPF), which will be published alongside your paper.
- 4) A complete author checklist, which you can download from our author guidelines (<https://www.embopress.org/page/journal/17574684/authorguide#submissionofrevisions>). Please insert information in the checklist that is also reflected in the manuscript. The completed author checklist will also be part of the RPF.
- 5) Please note that all corresponding authors are required to supply an ORCID ID for their name upon submission of a revised manuscript.
- 6) It is mandatory to include a 'Data Availability' section after the Materials and Methods. Before submitting your revision, primary datasets produced in this study need to be deposited in an appropriate public database, and the accession numbers and

database listed under 'Data Availability'. Please remember to provide a reviewer password if the datasets are not yet public (see <https://www.embopress.org/page/journal/17574684/authorguide#dataavailability>).

12) Author contributions: You will be asked to provide CRediT (Contributor Role Taxonomy) terms in the submission system. These replace a narrative author contribution section in the manuscript.

13) A Conflict of Interest statement should be provided in the main text.

14) Every published paper now includes a 'Synopsis' to further enhance discoverability. Synopses are displayed on the journal webpage and are freely accessible to all readers. They include a short stand first (maximum of 300 characters, including space) as well as 2-5 one-sentences bullet points that summarizes the paper. Please write the bullet points to summarize the key NEW findings. They should be designed to be complementary to the abstract - i.e. not repeat the same text. We encourage inclusion of key acronyms and quantitative information (maximum of 30 words / bullet point). Please use the passive voice. Please attach

these in a separate file or send them by email, we will incorporate them accordingly.

15) Include a Reagents and Tools Table as part of the Methods section, which can be downloaded from our author guidelines (<https://www.embopress.org/page/journal/17574684/authorguide#structuredmethods>)

***** Reviewer's comments *****

Referee #1 (Remarks for Author):

The manuscript from Liu and colleagues tackles one of the quality issues associated with recombinant AAV production i.e. the presence of bacterial DNA as contaminants. Although this issue can be greatly minimized by optimizing the production conditions and the plasmid material, I found the approach innovative and worth considering for publication after a thorough revision.

This reviewer has two main issues related to the major findings of this manuscript:

1. The authors insist on saying that their new plasmid configuration allows to avoid the asynchronous presence of pTrans and pCis and try to demonstrate that the formation of empty particles is due to the translation of pTrans in the absence of pCis. However, they performed a time course where they showed that at early time points (i.e. 24 hours) the three approaches had similar levels of expression of both Cap protein and mRNA (Figure 3F). This reviewer wonders if their enhanced encapsidation efficiency is simply due to the fact that their construct expresses lower amounts of Cap gene at later time points. Also, the authors based all of their conclusions on the use of a single transgene expression cassette bearing a constitutive promoter likely expressed in producer cell lines. Similar experiments have to be performed with promoters not expressed in cell lines, hAAT to cite one.

Finally, how can the asynchronous mechanism explain the known differences in full to empty ratios that can be observed by using different transgene expression cassettes with the triple transfection method?

2. The authors base all their claims on quality on the ratio between ELISA-measured capsids and ddPCR-measured genomes. A thorough characterization of the vector produced in Figure 5 is needed to support their claims. In particular, sequencing for evaluating the presence of other plasmid contaminants and an AUC to exclude the formation of capsids containing intermediate forms of DNA that cannot be distinguished from the complete cassette based on their method. Finally, given the inclusion of ITRs in a plasmid containing Rep and Cap, an evaluation of the presence of wild-type AAV (i.e. including rep and/or cap genes) is very important.

Referee #2 (Remarks for Author):

Advancements in AAV manufacturing is critical to expanding the availability and safety of future gene therapies. This manuscript provides a novel method for the production of high quality AAV vectors and adequately explains how this method can be adapted to both pilot and commercial manufacturing. However, there are aspects of the resulting vector that still need to be explored.

While ddPCR was used to look at AmpR in all iterations of AAVPure[^]mfg, other common contaminants such as rep/cap and Helper genes were only reported in figure 5 using the 2.0 system in suspension. I believe it is important to assess these contaminants in the other systems reported. In addition, while the bxb1 transcript size is >7kb in this context, the authors did note to reduce burden for pHelper-Bxb1 a smaller Bxb1 spike-in could be used which may be packaged in an AAV vector. Were any ddPCR experiments conducted to detect Bxb1 packaging? Have you considered sending some of these preparations to next gen sequencing to determine if the formation of the minicircle alters genome homogeneity compared to traditional triple transfection? NGS will also be able to determine if any other backbone components are being preferentially packaged into the AAVs that are not included in the ddPCR panel.

The only construct used in this paper was an ITR flanked CB6-EGFP-pA with an expected size of ~2.3kb. Presumably this is a ssAAV genome. Do you expect a larger genome closer to the packaging limit of ~4.5-4.7kb to have the same efficiency as this relatively small genome? Packaging limitations are a common hurdle when selecting a transgene, it would be beneficial to show larger constructs can be used with this method. Have you tested this method with a scAAV genome?

Currently, the primary readout of this manuscript is ddPCR and ELISA to determine the %Full ratio. Have any of these

preparations been analyzed using AUC to determine the presence or absence of partial or overfilled capsids given the relatively small genome?

Overall, this manuscript provides an excellent path forward for AAV manufacturing, and with a few more analytical assays, will be very valuable to the field. While I still have questions, the method itself and data presented is extremely promising.

Referee #3 (Remarks for Author):

Manuscript EMM-2024-21046-V2-Q

High-purity AAV vector production utilizing recombined minicircle formation and genetic coupling

December 24, 2024

Triple plasmid transfection of HEK293 cells is the most widely used upstream method for the manufacture of rAAV vectors. Over the past 10 years, the transfection process has been successfully scaled to 2000L. However, little investment and advancement has been made in plasmid design to improve manufacturability and packaging. The AAVPureMfg plasmid design was demonstrated to show a 2-3 fold increase in genome-containing capsids (serotypes 2, 8 and 9) along with significantly reducing the plasmid prokaryotic DNA sequences packaged in the capsid. This was achieved by Bxb1-mediated excision of the transgene cassette (located within rep/cap) producing a minicircle cis construct that lacks bacterial sequences and ensures synchronized expression of only colocalized trans and cis constructs in productive cells. In addition, rep/cap is only functional in the presence of Ad helper plasmid containing Bxb1 preventing excess empty capsids being produced in the absence of ITR genomes. Yields are comparable to the standard triple plasmid transfection process. This design is a significant step forward in the manufacture of rAAV vectors using transfection and can be performed using 2 or 3 plasmids. The design is also compatible with a number of stable monoclonal cell line variants in the future for single plasmid transfection or to eliminate the need for transfection.

Minor comment #1:

This manuscript shows proof of concept that the AAVPureMfg plasmid design manufactures rAAV vector with comparable yields to the standard triple transfection process, significantly reduces the residual plasmid backbone DNA packaged and comparable packaging of a 2.3kb genome.

- Reviewer would recommend packaging a near-wt size ss and sc genomes and compare packaging (Full:partial:empty) via AUC and/or CD-MS to standard triple transfection. There may be an improvement in F:P:E (specifically F and P) using the AAVPureMfg
- Testing for rcAAV and comparing against standard triple transfection
- Comparing rescue out of and subsequent genome replication between standard ITR plasmid format and AAVPUREMFG plasmid format (minicircle).

Minor comment #2:

For AAVPureMfg 3.0 stable cell line, reviewer recommends testing for residual Puromycin

Minor Comment #3:

Transfection DoEs could be performed to further improve yields and F:P:E packaging (see comment 1) using the AAVPureMfg plasmids.

Minor Comment #4:

Reviewer acknowledges that the experiments performed at 30 mL scale in suspension Expi293 cells will scale to larger volumes based on transfection scalability shown by several gene therapy companies and CMOs. It would be valuable in this manuscript or a future manuscript using AAVPureMfg to perform two 3L to 10L bioreactor manufacturing runs with an industry standard downstream purification process (chromatography with or without iodixanol density gradient) to generate enough rAAV vector material to perform standard QC release testing (including residuals, infectivity/potency, rcAAV) to compare against triple plasmid transfection.

Overall, the manuscript is technically sound, the experiments conducted along with their respective figures are detailed and graphically presented well and the conclusions reached based on the data are accurate. The reviewer's comments are minor and for the authors to consider for future experiments as they continue to validate and improve the AAVPUREMFG technology. The reviewer recommends accepting this manuscript for publication as the authors have demonstrated POC that will ultimately improve rAAV vector manufacturing.

Referee #1 (Remarks for Author):

The manuscript from Liu and colleagues tackles one of the quality issues associated with recombinant AAV production i.e. the presence of bacterial DNA as contaminants. Although this issue can be greatly minimized by optimizing the production conditions and the plasmid material, I found the approach innovative and worth considering for publication after a thorough revision.

Response 1: We thank the reviewer for commenting on the novelty of our study.

This reviewer has two main issues related to the major findings of this manuscript:

1. The authors insist on saying that their new plasmid configuration allows to avoid the asynchronous presence of pTrans and pCis and try to demonstrate that the formation of empty particles is due to the translation of pTrans in the absence of pCis. However, they performed a time course where they showed that at early time points (i.e. 24 hours) the three approaches had similar levels of expression of both Cap protein and mRNA (Figure 3F). This reviewer wonders if their enhanced encapsidation efficiency is simply due to the fact that their construct expresses lower amounts of Cap gene at later time points.

Response 1-1a: We greatly appreciate this alternative explanation to our observation. To test whether lower Cap expression can improve full capsid ratio, we devised an inducible Cap expression system with the cumate gene-switch (Mullick A et al. *BMC Biotechnol* 2006, PMID: 17083727). When used for rAAV production, it expressed Cap when cumate was added to the culture media in a dose-dependent manner. Consistent with the Reviewer's prediction, as Cap expression was reduced, the full capsid ratio indeed moderately increased. However, this was accompanied by a marked reduction in the rAAV genome titer (i.e., full capsid). By contrast, AAVPure^{Mfg} improved the full capsid ratio without compromising genome titer. Thus, we concluded that the improved full capsid ratio with AAVPure^{Mfg} was not simply due to the fact that it expressed lower amounts of Cap gene at all time points. This set of new data and analysis is now included in the revision (**Figure EV2, lines 216-222**).

Also, the authors based all of their conclusions on the use of a single transgene expression cassette bearing a constitutive promoter likely expressed in producer cell lines. Similar experiments have to be performed with promoters not expressed in cell lines, hAAT to cite one.

Response 1-1b: We agree and have tested AAVPure^{Mfg} with two additional transgene cassettes: one with a large transgene driven by the human rhodopsin kinase (hRK) promoter (a photoreceptor-specific promoter commonly used in AAV-mediated ocular gene therapy), and the other in a self-complementary (sc) genome configuration. In both cases, AAVPure^{Mfg} consistently showed improved full capsid ratio and reduced plasmid backbone DNA encapsidation (**Figure EV3C-F, lines 239-241**).

Finally, how can the asynchronous mechanism explain the known differences in full to empty ratios that can be observed by using different transgene expression cassettes with the triple transfection method?

Response 1-1c: We agree with the Reviewer that the transgene cassette can influence full capsid ratio. The asynchronous mechanism proposed in this study does not explain such an influence. We postulate that expression of certain transgenes may alter the cellular environment, which ultimately impacts rAAV production. Additionally, the ITR-flanked DNA sequence may play a direct role in determining packaging efficiency. We have included these discussion in the revision (**lines 344-348**).

2. The authors base all their claims on quality on the ratio between ELISA-measured capsids and

ddPCR-measured genomes. A thorough characterization of the vector produced in Figure 5 is needed to support their claims. In particular, sequencing for evaluating the presence of other plasmid contaminants and an AUC to exclude the formation of capsids containing intermediate forms of DNA that cannot be distinguished from the complete cassette based on their method. Finally, given the inclusion of ITRs in a plasmid containing Rep and Cap, an evaluation of the presence of wild-type AAV (i.e. including rep and/or cap genes) is very important.

Response 1-2: We thank the reviewer for the suggestion to thoroughly characterize vector attributes with advanced techniques. We performed single-molecule, long-read DNA sequencing (PacBio) of encapsidated vector DNA. Consistent with the ddPCR results, PacBio sequencing showed that the plasmid backbone DNA contamination in AAVPure^{Mfg} was reduced by 30-fold as compared to triple transfection. Host cell DNA and pHelper DNA levels were comparable between the two platforms. Regarding wild-type AAV (wtAAV), we detected a single read of Rep-Cap out of approximately 600,000 reads in each sample, suggesting low and comparable levels of potential wtAAV across both platforms. These data are included in the revision (**Table EV2, lines 185-191**).

We acknowledge that AUC is the industry gold standard for quantifying full, intermediate, and empty capsids. However, it requires a large amount of purified vectors (typically 450 uL at 1e13 vp/mL), which is not suitable for screening various AAVPure^{Mfg} conditions in our study. To our knowledge, ddPCR/ELISA remains the only method available to date for quantifying full capsid ratio in crude lysate samples, and it is commonly used by many laboratories across industry and academia (e.g., PMID: 35256603, 37096037, 39941089). Nonetheless, for a representative comparison between AAVPure^{Mfg} and triple transfection, we tried to set up affinity chromatography purification (as our established density gradient purification method separates full from empty capsids). Additionally, we were collaborating with a colleague to establish an AUC protocol. Besides the technical difficulties, we were also facing nationwide research budget cuts and recently imposed institutional spending constraints, which further complicated the resource-intensive tasks of large-scale rAAV production, chromatography purification, and AUC characterization. Therefore, despite our best efforts, we were unable to provide AUC data to address the limitation associated with the ddPCR/ELISA method. At the editor's discretion, we provide the detailed explanation as outlined above, and have discussed this limitation in the revision (**lines 351-357**).

Referee #2 (Remarks for Author):

Advancements in AAV manufacturing is critical to expanding the availability and safety of future gene therapies. This manuscript provides a novel method for the production of high quality AAV vectors and adequately explains how this method can be adapted to both pilot and commercial manufacturing. However, there are aspects of the resulting vector that still need to be explored.

Response 2: We thank the reviewer for commenting on the novelty of our method.

While ddPCR was used to look at AmpR in all iterations of AAVPure^{mfg}, other common contaminants such as rep/cap and Helper genes were only reported in figure 5 using the 2.0 system in suspension. I believe it is important to assess these contaminants in the other systems reported. In addition, while the bxb1 transcript size is >7kb in this context, the authors did note to reduce burden for pHelper-Bxb1 a smaller Bxb1 spike-in could be used which may be packaged in an AAV vector. Were any ddPCR experiments conducted to detect Bxb1 packaging? Have you considered sending some of these preparations to next gen sequencing to determine if the formation of the minicircle alters genome homogeneity compared to traditional triple transfection? NGS will also be able to determine if any other

backbone components are being preferentially packaged into the AAVs that are not included in the ddPCR panel.

Response 2-1: We thank the reviewer for the suggestion to thoroughly characterize vector attributes with advanced techniques. We performed single-molecule, long-read DNA sequencing (PacBio) of encapsidated vector DNA. Consistent with the ddPCR results, PacBio sequencing showed that the plasmid backbone DNA contamination in AAVPure^{Mfg} was reduced by 30-fold as compared to triple transfection. Host cell DNA and pHelper DNA levels were comparable between the two platforms. Regarding replication-competent AAV (rcAAV), we detected a single read of Rep-Cap out of approximately 600,000 reads in each sample, suggesting low and comparable levels of potential rcAAV across both platforms. These data are included in the revision (**Table EV2, lines 185-191**).

Additionally, we designed a high-sensitivity, targeted ddPCR assay to quantify potential Bxb1 contaminants in purified AAV vectors produced with AAVPure^{Mfg} 2.0. This analysis showed that any possible Bxb1 encapsidation would be below the detection limit of ddPCR. These data have been included in the revision (**Appendix Figure S7, lines 340-341**).

Regarding vector genome homogeneity, we performed denaturing alkaline gel electrophoresis and found that AAVPure^{Mfg} seems to improve vector genome homogeneity by mitigating double-genome rAAV packaging (**Figure 5E, S8**). We provided plausible explanations in the Discussion section (**lines 358-363**).

The only construct used in this paper was an ITR flanked CB6-EGFP-pA with an expected size of ~2.3kb. Presumably this is a ssAAV genome. Do you expect a larger genome closer to the packaging limit of ~4.5-4.7kb to have the same efficiency as this relatively small genome? Packaging limitations are a common hurdle when selecting a transgene, it would be beneficial to show larger constructs can be used with this method. Have you tested this method with a scAAV genome?

Response 2-2: We agree and have tested AAVPure^{Mfg} with two additional transgene cassettes: one with a large transgene cassette (4.7 kb), and the other in a self-complementary (sc) genome configuration. In both cases, AAVPure^{Mfg} consistently showed improved full capsid ratio and reduced plasmid backbone DNA encapsidation (**Figure EV3, lines 239-241**).

Currently, the primary readout of this manuscript is ddPCR and ELISA to determine the %Full ratio. Have any of these preparations been analyzed using AUC to determine the presence or absence of partial or overfilled capsids given the relatively small genome?

Response 2-3: We acknowledge that AUC is the industry gold standard for quantifying overfilled, full, partial, and empty capsids. However, it requires a large amount of purified vectors (typically 450 uL at 1e13 vp/mL), which is not suitable for screening various AAVPure^{Mfg} conditions in our study. To our knowledge, ddPCR/ELISA remains the only method available to date for quantifying full capsid ratio in crude lysate samples, and it is commonly used by many laboratories across industry and academia (e.g., PMID: 35256603, 37096037, 39941089). Nonetheless, for a representative comparison between AAVPure^{Mfg} and triple transfection, we tried to set up affinity chromatography purification (as our established density gradient purification method separates full from empty capsids). Additionally, we were collaborating with a colleague to establish an AUC protocol. Besides the technical difficulties, we were also facing nation-wide research budget cuts and recently imposed institutional spending constraints, which further complicated the resource-intensive tasks of large-scale rAAV production, chromatography purification, and AUC characterization. Therefore, despite our best efforts, we were unable to provide AUC data to address the limitation associated with the ddPCR/ELISA method. At the editor's discretion, we provide our detailed explanation as outlined above, and have discussed this limitation in the revision (**lines 351-357**).

Overall, this manuscript provides an excellent path forward for AAV manufacturing, and with a few more analytical assays, will be very valuable to the field. While I still have questions, the method itself and data presented is extremely promising.

Response 2-4: We greatly appreciate the reviewer's encouraging comment on this study.

Referee #3 (Remarks for Author):

Manuscript EMM-2024-21046-V2-Q

High-purity AAV vector production utilizing recombined minicircle formation and genetic coupling

December 24, 2024

Triple plasmid transfection of HEK293 cells is the most widely used upstream method for the manufacture of rAAV vectors. Over the past 10 years, the transfection process has been successfully scaled to 2000L. However, little investment and advancement has been made in plasmid design to improve manufacturability and packaging. The AAVPureMfg plasmid design was demonstrated to show a 2-3 fold increase in genome-containing capsids (serotypes 2, 8 and 9) along with significantly reducing the plasmid prokaryotic DNA sequences packaged in the capsid. This was achieved by Bxb1-mediated excision of the transgene cassette (located within rep/cap) producing a minicircle cis construct that lacks bacterial sequences and ensures synchronized expression of only colocalized trans and cis constructs in productive cells. In addition, rep/cap is only functional in the presence of Ad helper plasmid containing Bxb1 preventing excess empty capsids being produced in the absence of ITR genomes. Yields are comparable to the standard triple plasmid transfection process. This design is a significant step forward in the manufacture of rAAV vectors using transfection and can be performed using 2 or 3 plasmids. The design is also compatible with a number of stable monoclonal cell line variants in the future for single plasmid transfection or to eliminate the need for transfection.

Response 3: We thank the reviewer for commenting on the significance of our study.

Minor comment #1:

This manuscript shows proof of concept that the AAVPureMfg plasmid design manufactures rAAV vector with comparable yields to the standard triple transfection process, significantly reduces the residual plasmid backbone DNA packaged and comparable packaging of a 2.3kb genome.

- Reviewer would recommend packaging a near-wt size ss and sc genomes and compare packaging (Full:partial:empty) via AUC and/or CD-MS to standard triple transfection. There may be an improvement in F:P:E (specifically F and P) using the AAVPureMfg

Response 3-1a: We fully agree with the reviewer, and have tested AAVPure^{Mfg} with two additional transgene cassettes: one with a large transgene cassette (4.7 kb), and the other in a self-complementary (sc) genome configuration. In both cases, AAVPure^{Mfg} consistently showed improved full capsid ratio and reduced plasmid backbone DNA encapsidation (**Figure EV3, lines 239-241**).

We acknowledge that AUC is the industry gold standard for quantifying full, intermediate, and empty capsids. However, it requires a large amount of purified vectors (typically 450 uL at 1e13 vp/mL), which is not suitable for screening various AAVPure^{Mfg} conditions in our study. To our knowledge, ddPCR/ELISA remains the only method available to date for quantifying full capsid ratio in crude lysate samples, and it is commonly used by many laboratories across industry and academia (e.g., PMID: 35256603, 37096037, 39941089). Nonetheless, for a representative comparison between AAVPure^{Mfg}

and triple transfection, we tried to set up affinity chromatography purification (as our established density gradient purification method separates full from empty capsids). Additionally, we were collaborating with a colleague to establish an AUC protocol. Besides the technical difficulties, we were also facing nationwide research budget cuts and recently imposed institutional spending constraints, which further complicated the resource-intensive tasks of large-scale rAAV production, chromatography purification, and AUC characterization. Therefore, despite our best efforts, we were unable to provide AUC data to address the limitation associated with the ddPCR/ELISA method. At the editor's discretion, we provide our detailed explanation as outlined above, and have discussed this limitation in the revision (**lines 351-357**).

- Testing for rcAAV and comparing against standard triple transfection

Response 3-1b: We performed single-molecule, long-read DNA sequencing (PacBio) of encapsidated vector DNA to further characterize DNA impurities. Regarding replication-competent AAV (rcAAV), we detected a single read of Rep-Cap out of approximately 600,000 reads in each sample, suggesting low and comparable levels of potential rcAAV across both platforms. These data are included in the revision (**Table EV2, lines 189-191**).

- Comparing rescue out of and subsequent genome replication between standard ITR plasmid format and AAVPUREMFG plasmid format (minicircle).

Response 3-1c: We thank the reviewer for the suggestion to compare vector genome rescue and replication. We extracted total cellular DNA from cells undergoing either triple transfection or AAVPure^{Mfg} 2.0, followed by DpnI treatment to remove residual plasmids before vector genome quantification. This is a well-established procedure to quantify viral DNA in the presence of interfering plasmid DNA (e.g., PMID: 19211760). ddPCR detected similar levels of DpnI-resistant vector genome DNA between the two platforms, suggesting comparable rescue/replication efficiency between standard ITR plasmid format and AAVPure^{Mfg} plasmid format (minicircle). This set of data have been included in the revision (**Appendix Figure S5, lines 245-248**).

Minor comment #2:

For AAVPureMfg 3.0 stable cell line, reviewer recommends testing for residual Puromycin

Response 3-2: We agree and have designed a high-sensitivity, targeted ddPCR assay to quantify potential Puromycin contaminants in purified AAV vectors produced with AAVPure^{Mfg} 3.0. This analysis showed that any possible Puromycin gene encapsidation would be below the detection limit of ddPCR. These data have been included in the revision (**Appendix Figure S7, lines 340-341**).

Minor Comment #3:

Transfection DoEs could be performed to further improve yields and F:P:E packaging (see comment 1) using the AAVPureMfg plasmids.

Response 3-3: We agree with the reviewer that a DoE approach could be used to systematically optimize transfection variables for higher rAAV production yield and quality. As a proof-of-concept, we screened multiple conditions in the new experiments to package a large transgene or a sc genome, with a focus on the two key plasmids used in AAVPure^{Mfg} (**Figure EV3C-F**). Although it was not a comprehensive DoE approach, the data demonstrated the potential of optimizing transfection conditions to maximize yield. Additionally, it suggested that effective optimization, such as using DoE, could be performed in the context of specific transgene and genome configuration. While we respectfully believe that conducting a systematic DoE experiment with the EGFP model construct would

be beyond the scope of this study, we have discussed the potential integration of a DoE approach into AAVPure^{Mfg} (lines 348-351).

Minor Comment #4:

Reviewer acknowledges that the experiments performed at 30 mL scale in suspension Expi293 cells will scale to larger volumes based on transfection scalability shown by several gene therapy companies and CMOs. It would be valuable in this manuscript or a future manuscript using AAVPureMfg to perform two 3L to 10L bioreactor manufacturing runs with an industry standard downstream purification process (chromatography with or without iodixanol density gradient) to generate enough rAAV vector material to perform standard QC release testing (including residuals, infectivity/potency, rcAAV) to compare against triple plasmid transfection.

Response 3-4: We fully agree with the reviewer that a comparative study with industry-relevant production scale, upstream process, downstream process, and analytical release testing would strengthen the utility of AAVPure^{Mfg}. We are actively seeking collaboration with an industry partner and will conduct such a study when opportunities arise. This direction is added in the revision (lines 356-357).

Overall, the manuscript is technically sound, the experiments conducted along with their respective figures are detailed and graphically presented well and the conclusions reached based on the data are accurate. The reviewer's comments are minor and for the authors to consider for future experiments as they continue to validate and improve the AAVPUREMFG technology. The reviewer recommends accepting this manuscript for publication as the authors have demonstrated POC that will ultimately improve rAAV vector manufacturing.

Response 3-5: We thank the reviewer for the thoughtful comments and suggestions on the future development of this technology.

12th Apr 2025

Dear Dr. Wang,

Thank you for the submission of your revised manuscript to EMBO Molecular Medicine. We have now received the enclosed reports from the referees that were asked to re-assess it. I am pleased to inform you that we will be able to accept your manuscript pending the following final amendments:

1) Please address referee #1 concern about AUC by discussion.

2) Figures and Tables:

- EV figures and EV tables should be uploaded as individual files. The EV figure legends should be removed from the figures and only remain in the main manuscript text, under the heading "Expanded View Figure Legends"

- We note that some panels are reused e.g. Figure 5E and Appendix S6. Please cite in the respective figure legend every reused panel.

3) In the main manuscript file, please do the following:

- Please address all comments suggested by our data editors listed below:

o Data availability statement:

1. Please note that the specific URL for PRJNA1239179 dataset is not provided in the data availability statement.

o Figure legends:

1. Please note that the exact p values are not provided in the legends of figures 2G, 3C, D, F; 4C, D, E, I, J; 5F, EV1 B, C; EV2 C, EV3 B, D, F; EV4 E.

2. Please note that information related to n is missing in the legends of figures 1E, 2C, E, F, G; 3B, C, D, F; 4B, C, D, E, G, H, I, J; 5B-D, F-I; EV1 B, C; EV2 C, D, E, F; EV3 B, D, F; EV4 B-E.

3. Please note that the error bars are not defined in the legends of figures EV2 C, D, E, F.

4. Please note that scale bar and its definition are missing for figures 2D, EV5 E.

- Correct order and headings of manuscript sections: Abstract / Keywords / The Paper Explained / Introduction / Results / Discussion / Methods / Data Availability / Acknowledgements / Disclosure and Competing Interests Statement / References / Main Figure Legends / Tables / Expanded View Figure Legends.

- Add callouts for Table EV3 and individual panels of Figure EV3. Also, callouts for Appendix tables should be corrected to "Appendix Table S1" etc. and they should be called out in a sequential order. We also note that Appendix tables are missing in the Appendix file. Please correct.

- Author contributions: Please remove it from the manuscript and specify author contributions in our submission system. CRediT has replaced the traditional author contributions section because it offers a systematic machine-readable author contributions format that allows for more effective research assessment. You are encouraged to use the free text boxes beneath each contributing author's name to add specific details on the author's contribution. More information is available in our guide to authors:

<https://www.embopress.org/page/journal/17574684/authorguide#authorshipguidelines>

- Indicate in legends exact n and exact p values, not a range, along with the statistical test used. To keep the figures "clear" some authors found providing an Appendix table Sx with all exact p-values preferable. You are welcome to do this if you want to.

- BioRender information should be removed from Acknowledgements and entered in a dedicated section in the Methods using the following format:

Graphics:

(some of the... OR Figure #... OR synopsis) Graphics were created with BioRender.com.

- In Methods, statistical paragraph should reflect all information that you have filled in the Authors Checklist, especially regarding randomization, blinding, replication.

- Please include structured Methods section that includes a Reagents and Tools Table (should be uploaded as a separate file) followed by a Methods and Protocols section. More information on how to adhere to this format as well as downloadable templates (.docx) for the Reagents and Tools Table can be found in our author guidelines:

<https://www.embopress.org/page/journal/17574684/authorguide#structuredmethods>

An example of a paper with Structured Methods can be found here:

<https://www.embopress.org/doi/full/10.1038/s44320-024-00037-6#sec-4>

- In data availability please use the following format to report the accession number of your data:

[data type]: [full name of the resource] [accession number/identifier] ([doi or URL or identifiers.org/DATABASE:ACCESSION])

Please check "Author Guidelines" for more information.

<https://www.embopress.org/page/journal/17574684/authorguide#availabilityofpublishedmaterial>

4) Appendix: Please add missing tables and name them "Appendix Table S1" etc., place them after Appendix Figures and update their callouts in the main manuscript file. Also add page numbers to the table of content.

5) The Paper Explained: Please add it to the main manuscript file.

6) Synopsis:

- Synopsis image: Please resize the image to 550 px-wide x (300-600)-px high and upload it as a high-resolution jpeg file.
- Please check your synopsis text and image before submission with your revised manuscript. Please be aware that in the proof stage minor corrections only are allowed (e.g., typos).

7) As part of the EMBO Publications transparent editorial process initiative (see our Editorial at <http://embomolmed.embopress.org/content/2/9/329>), EMBO Molecular Medicine will publish online a Review Process File (RPF) to accompany accepted manuscripts. This file will be published in conjunction with your paper and will include the anonymous referee reports, your point-by-point response and all pertinent correspondence relating to the manuscript. Let us know whether you agree with the publication of the RPF and as here, if you want to remove or not any figures from it prior to publication. Please note that the Authors checklist will be published at the end of the RPF.

8) Please provide a point-by-point letter INCLUDING my comments as well as the reviewer's reports and your detailed responses (as Word file).

I look forward to reading a new revised version of your manuscript as soon as possible.

Yours sincerely,

Zeljko Durdevic

*** Instructions to submit your revised manuscript ***

- 1) a .docx formatted version of the manuscript text (including Figure legends and tables)
- 2) Separate figure files*
- 3) supplemental information as Expanded View and/or Appendix. Please carefully check the authors guidelines for formatting Expanded view and Appendix figures and tables at <https://www.embopress.org/page/journal/17574684/authorguide#expandedview>
- 4) a letter INCLUDING the reviewer's reports and your detailed responses to their comments (as Word file).
- 5) The paper explained: EMBO Molecular Medicine articles are accompanied by a summary of the articles to emphasize the major findings in the paper and their medical implications for the non-specialist reader. Please provide a draft summary of your article highlighting
 - the medical issue you are addressing,
 - the results obtained and
 - their clinical impact.

6) Author contributions: the contribution of every author must be detailed in a separate section.

7) EMBO Molecular Medicine now requires a complete author checklist (<https://www.embopress.org/page/journal/17574684/authorguide>) to be submitted with all revised manuscripts. Please use the checklist as guideline for the sort of information we need WITHIN the manuscript. The checklist should only be filled with page numbers where the information can be found. This is particularly important for animal reporting, antibody dilutions (missing) and exact values and n that should be indicated instead of a range.

8) Every published paper now includes a 'Synopsis' to further enhance discoverability. Synopses are displayed on the journal webpage and are freely accessible to all readers. They include a short stand first (maximum of 300 characters, including space) as well as 2-5 one sentence bullet points that summarise the paper. Please write the bullet points to summarise the key NEW findings. They should be designed to be complementary to the abstract - i.e. not repeat the same text. We encourage inclusion of key acronyms and quantitative information (maximum of 30 words / bullet point). Please use the passive voice. Please attach these in a separate file or send them by email, we will incorporate them accordingly.

You are also welcome to suggest a striking image or visual abstract to illustrate your article. If you do please provide a jpeg file 550 px-wide x 300-600px high.

9) A Conflict of Interest statement should be provided in the main text

10) Please note that we now mandate that all corresponding authors list an ORCID digital identifier. This takes <90 seconds to complete. We encourage all authors to supply an ORCID identifier, which will be linked to their name for unambiguous name identification.

Currently, our records indicate that the ORCID for your account is 0000-0001-9079-2360.

Link Not Available

11) Include a Reagents and Tools Table as part of the Methods section, which can be downloaded from our author guidelines (<https://www.embopress.org/page/journal/17574684/authorguide#structuredmethods>)

Photos 400-800 DPI

*Additional important information regarding figures and illustrations can be found at <https://bit.ly/EMBOPressFigurePreparationGuideline>. See also figure legend preparation guidelines: <https://www.embopress.org/page/journal/17574684/authorguide#figureformat>

***** Reviewer's comments *****

Referee #1 (Comments on Novelty/Model System for Author):

There is no model organism used in this manuscript. The only comment I have in terms of adequacy is the lack of the use of a different technique to support the main conclusion of the manuscript. All the results related to empty/full ratio evaluation are obtained by dd-PCR and ELISA. This may seem trivial but perhaps the method leads to increased fragmentation of the recombinant, encapsidated genome that cannot be estimated with the current method. The use of analytical ultracentrifugation was suggested by the three reviewers but the authors have not provided any data in this direction.

Referee #1 (Remarks for Author):

The authors addressed most of my concerns, except the most relevant one raised by the two other reviewers. To appreciate the value of this work, it is fundamental to include an AUC estimation of the quality of the vectors. The use of dd-PCR coupled with ELISA greatly underestimates the presence of fragmented genomes or partially filled AAV vectors. This is an underestimated issue in gene therapy that deserves evaluation.

Referee #2 (Comments on Novelty/Model System for Author):

In my view, the authors did an adequate job of addressing each point made by myself and the other reviewers. As such, I gave the technical quality of the manuscript a high score to reflect the rigor used in testing triple transfection vs the AAVPure system.

Rather than simply using a minicircle to reduce plasmid cross-packaging the authors used a novel system to produce "mini-circles" following transfection only in cells receiving both the pHelp and pTrans/cis plasmids. The approach is very novel and has the added benefit of producing a high degree of full capsids.

Together, this advancement in AAV manufacturing has a high degree of medical significance by easing some of the hurdles present in AAV production. Namely, increasing the safety profile by reducing plasmid cross-packing and reducing the quantity of empty capsids that need to be cleared during purification.

The model system used was adequate as AAV manufacturing is predominantly conducted in HEK293 cells.

Referee #2 (Remarks for Author):

I thank the authors for addressing all of my comments during initial review. I believe they have adequately addressed all of my concerns. While I would still like to see AUC data, I understand that the method is not readily available. It is my hope in the future as the production system is more widely applied AUC can be conducted to create a clearer picture of Empty vs Partial vs Full vs Overfilled AAVs being produced. Overall, I look forward to the finalized publication and how this new method may be applied to AAV therapies moving forward.

Point-by-point Response

Referee #1 (Remarks for Author):

The manuscript from Liu and colleagues tackles one of the quality issues associated with recombinant AAV production i.e. the presence of bacterial DNA as contaminants. Although this issue can be greatly minimized by optimizing the production conditions and the plasmid material, I found the approach innovative and worth considering for publication after a thorough revision.

Response 1: We thank the reviewer for commenting on the novelty of our study.

This reviewer has two main issues related to the major findings of this manuscript:

1. The authors insist on saying that their new plasmid configuration allows to avoid the asynchronous presence of pTrans and pCis and try to demonstrate that the formation of empty particles is due to the translation of pTrans in the absence of pCis. However, they performed a time course where they showed that at early time points (i.e. 24 hours) the three approaches had similar levels of expression of both Cap protein and mRNA (Figure 3F). This reviewer wonders if their enhanced encapsidation efficiency is simply due to the fact that their construct expresses lower amounts of Cap gene at later time points.

Response 1-1a: We greatly appreciate this alternative explanation to our observation. To test whether lower Cap expression can improve full capsid ratio, we devised an inducible Cap expression system with the cumate gene-switch (Mullick A et al. *BMC Biotechnol* 2006, PMID: 17083727). When used for rAAV production, it expressed Cap when cumate was added to the culture media in a dose-dependent manner. Consistent with the Reviewer's prediction, as Cap expression was reduced, the full capsid ratio indeed moderately increased. However, this was accompanied by a marked reduction in the rAAV genome titer (i.e., full capsid). By contrast, AAVPure^{Mfg} improved the full capsid ratio without compromising genome titer. Thus, we concluded that the improved full capsid ratio with AAVPure^{Mfg} was not simply due to the fact that it expressed lower amounts of Cap gene at all time points. This set of new data and analysis is now included in the revision (**Figure EV2, lines 216-222**).

Also, the authors based all of their conclusions on the use of a single transgene expression cassette bearing a constitutive promoter likely expressed in producer cell lines. Similar experiments have to be performed with promoters not expressed in cell lines, hAAT to cite one.

Response 1-1b: We agree and have tested AAVPure^{Mfg} with two additional transgene cassettes: one with a large transgene driven by the human rhodopsin kinase (hRK) promoter (a photoreceptor-specific promoter commonly used in AAV-mediated ocular gene therapy), and the other in a self-complementary (sc) genome configuration. In both cases, AAVPure^{Mfg} consistently showed improved full capsid ratio and reduced plasmid backbone DNA encapsidation (**Figure EV3C-F, lines 239-241**).

Finally, how can the asynchronous mechanism explain the known differences in full to empty ratios that can be observed by using different transgene expression cassettes with the triple transfection method?

Response 1-1c: We agree with the Reviewer that the transgene cassette can influence full capsid ratio. The asynchronous mechanism proposed in this study does not explain such an influence. We postulate that expression of certain transgenes may alter the cellular environment, which ultimately impacts rAAV production. Additionally, the ITR-flanked DNA sequence may play a direct role in determining packaging efficiency. We have included these discussion in the revision (**lines 344-348**).

2. The authors base all their claims on quality on the ratio between ELISA-measured capsids and ddPCR-measured genomes. A thorough characterization of the vector produced in Figure 5 is needed to support their claims. In particular, sequencing for evaluating the presence of other plasmid contaminants and an AUC to exclude the formation of capsids containing intermediate forms of DNA that cannot be distinguished from the complete cassette based on their method. Finally, given the inclusion of ITRs in a plasmid containing Rep and Cap, an evaluation of the presence of wild-type AAV (i.e. including rep and/or cap genes) is very important.

Response 1-2: We thank the reviewer for the suggestion to thoroughly characterize vector attributes with advanced techniques. We performed single-molecule, long-read DNA sequencing (PacBio) of encapsidated vector DNA. Consistent with the ddPCR results, PacBio sequencing showed that the plasmid backbone DNA contamination in AAVPure^{Mfg} was reduced by 30-fold as compared to triple transfection. Host cell DNA and pHelper DNA levels were comparable between the two platforms. Regarding wild-type AAV (wtAAV), we detected a single read of Rep-Cap out of approximately 600,000 reads in each sample, suggesting low and comparable levels of potential wtAAV across both platforms. These data are included in the revision (**Table EV2, lines 185-191**).

We acknowledge that AUC is the industry gold standard for quantifying full, intermediate, and empty capsids. However, it requires a large amount of purified vectors (typically 450 uL at 1e13 vp/mL), which is not suitable for screening various AAVPure^{Mfg} conditions in our study. To our knowledge, ddPCR/ELISA remains the only method available to date for quantifying full capsid ratio in crude lysate samples, and it is commonly used by many laboratories across industry and academia (e.g., PMID: 35256603, 37096037, 39941089). Nonetheless, for a representative comparison between AAVPure^{Mfg} and triple transfection, we tried to set up affinity chromatography purification (as our established density gradient purification method separates full from empty capsids). Additionally, we were collaborating with a colleague to establish an AUC protocol. Besides the technical difficulties, we were also facing nationwide research budget cuts and recently imposed institutional spending constraints, which further complicated the resource-intensive tasks of large-scale rAAV production, chromatography purification, and AUC characterization. Therefore, despite our best efforts, we were unable to provide AUC data to address the limitation associated with the ddPCR/ELISA method. At the editor's discretion, we provide the detailed explanation as outlined above, and have discussed this limitation in the revision (**lines 351-357**).

Referee #2 (Remarks for Author):

Advancements in AAV manufacturing is critical to expanding the availability and safety of future gene therapies. This manuscript provides a novel method for the production of high quality AAV vectors and adequately explains how this method can be adapted to both pilot and commercial manufacturing. However, there are aspects of the resulting vector that still need to be explored.

Response 2: We thank the reviewer for commenting on the novelty of our method.

While ddPCR was used to look at AmpR in all iterations of AAVPure^{mfg}, other common contaminants such as rep/cap and Helper genes were only reported in figure 5 using the 2.0 system in suspension. I believe it is important to assess these contaminants in the other systems reported. In addition, while the bxb1 transcript size is >7kb in this context, the authors did note to reduce burden for pHelper-Bxb1 a smaller Bxb1 spike-in could be used which may be packaged in an AAV vector. Were any ddPCR experiments conducted to detect Bxb1 packaging? Have you considered sending some of these

preparations to next gen sequencing to determine if the formation of the minicircle alters genome homogeneity compared to traditional triple transfection? NGS will also be able to determine if any other backbone components are being preferentially packaged into the AAVs that are not included in the ddPCR panel.

Response 2-1: We thank the reviewer for the suggestion to thoroughly characterize vector attributes with advanced techniques. We performed single-molecule, long-read DNA sequencing (PacBio) of encapsidated vector DNA. Consistent with the ddPCR results, PacBio sequencing showed that the plasmid backbone DNA contamination in AAVPure^{Mfg} was reduced by 30-fold as compared to triple transfection. Host cell DNA and pHelper DNA levels were comparable between the two platforms. Regarding replication-competent AAV (rcAAV), we detected a single read of Rep-Cap out of approximately 600,000 reads in each sample, suggesting low and comparable levels of potential rcAAV across both platforms. These data are included in the revision (**Table EV2, lines 185-191**).

Additionally, we designed a high-sensitivity, targeted ddPCR assay to quantify potential Bxb1 contaminants in purified AAV vectors produced with AAVPure^{Mfg} 2.0. This analysis showed that any possible Bxb1 encapsidation would be below the detection limit of ddPCR. These data have been included in the revision (**Appendix Figure S7, lines 340-341**).

Regarding vector genome homogeneity, we performed denaturing alkaline gel electrophoresis and found that AAVPure^{Mfg} seems to improve vector genome homogeneity by mitigating double-genome rAAV packaging (**Figure 5E, S8**). We provided plausible explanations in the Discussion section (**lines 358-363**).

The only construct used in this paper was an ITR flanked CB6-EGFP-pA with an expected size of ~2.3kb. Presumably this is a ssAAV genome. Do you expect a larger genome closer to the packaging limit of ~4.5-4.7kb to have the same efficiency as this relatively small genome? Packaging limitations are a common hurdle when selecting a transgene, it would be beneficial to show larger constructs can be used with this method. Have you tested this method with a scAAV genome?

Response 2-2: We agree and have tested AAVPure^{Mfg} with two additional transgene cassettes: one with a large transgene cassette (4.7 kb), and the other in a self-complementary (sc) genome configuration. In both cases, AAVPure^{Mfg} consistently showed improved full capsid ratio and reduced plasmid backbone DNA encapsidation (**Figure EV3, lines 239-241**).

Currently, the primary readout of this manuscript is ddPCR and ELISA to determine the %Full ratio. Have any of these preparations been analyzed using AUC to determine the presence or absence of partial or overfilled capsids given the relatively small genome?

Response 2-3: We acknowledge that AUC is the industry gold standard for quantifying overfilled, full, partial, and empty capsids. However, it requires a large amount of purified vectors (typically 450 uL at 1e13 vp/mL), which is not suitable for screening various AAVPure^{Mfg} conditions in our study. To our knowledge, ddPCR/ELISA remains the only method available to date for quantifying full capsid ratio in crude lysate samples, and it is commonly used by many laboratories across industry and academia (e.g., PMID: 35256603, 37096037, 39941089). Nonetheless, for a representative comparison between AAVPure^{Mfg} and triple transfection, we tried to set up affinity chromatography purification (as our established density gradient purification method separates full from empty capsids). Additionally, we were collaborating with a colleague to establish an AUC protocol. Besides the technical difficulties, we were also facing nation-wide research budget cuts and recently imposed institutional spending constraints, which further complicated the resource-intensive tasks of large-scale rAAV production, chromatography purification, and AUC characterization. Therefore, despite our best efforts, we were unable to provide AUC data to address the limitation associated with the ddPCR/ELISA

method. At the editor's discretion, we provide our detailed explanation as outlined above, and have discussed this limitation in the revision (lines 351-357).

Overall, this manuscript provides an excellent path forward for AAV manufacturing, and with a few more analytical assays, will be very valuable to the field. While I still have questions, the method itself and data presented is extremely promising.

Response 2-4: We greatly appreciate the reviewer's encouraging comment on this study.

Referee #3 (Remarks for Author):

Manuscript EMM-2024-21046-V2-Q

High-purity AAV vector production utilizing recombined minicircle formation and genetic coupling

December 24, 2024

Triple plasmid transfection of HEK293 cells is the most widely used upstream method for the manufacture of rAAV vectors. Over the past 10 years, the transfection process has been successfully scaled to 2000L. However, little investment and advancement has been made in plasmid design to improve manufacturability and packaging. The AAVPureMfg plasmid design was demonstrated to show a 2-3 fold increase in genome-containing capsids (serotypes 2, 8 and 9) along with significantly reducing the plasmid prokaryotic DNA sequences packaged in the capsid. This was achieved by Bxb1-mediated excision of the transgene cassette (located within rep/cap) producing a minicircle cis construct that lacks bacterial sequences and ensures synchronized expression of only colocalized trans and cis constructs in productive cells. In addition, rep/cap is only functional in the presence of Ad helper plasmid containing Bxb1 preventing excess empty capsids being produced in the absence of ITR genomes. Yields are comparable to the standard triple plasmid transfection process. This design is a significant step forward in the manufacture of rAAV vectors using transfection and can be performed using 2 or 3 plasmids. The design is also compatible with a number of stable monoclonal cell line variants in the future for single plasmid transfection or to eliminate the need for transfection.

Response 3: We thank the reviewer for commenting on the significance of our study.

Minor comment #1:

This manuscript shows proof of concept that the AAVPureMfg plasmid design manufactures rAAV vector with comparable yields to the standard triple transfection process, significantly reduces the residual plasmid backbone DNA packaged and comparable packaging of a 2.3kb genome.

- Reviewer would recommend packaging a near-wt size ss and sc genomes and compare packaging (Full:partial:empty) via AUC and/or CD-MS to standard triple transfection. There may be an improvement in F:P:E (specifically F and P) using the AAVPureMfg

Response 3-1a: We fully agree with the reviewer, and have tested AAVPure^{Mfg} with two additional transgene cassettes: one with a large transgene cassette (4.7 kb), and the other in a self-complementary (sc) genome configuration. In both cases, AAVPure^{Mfg} consistently showed improved full capsid ratio and reduced plasmid backbone DNA encapsidation (**Figure EV3, lines 239-241**).

We acknowledge that AUC is the industry gold standard for quantifying full, intermediate, and empty capsids. However, it requires a large amount of purified vectors (typically 450 uL at 1e13 vp/mL), which is not suitable for screening various AAVPure^{Mfg} conditions in our study. To our knowledge, ddPCR/ELISA remains the only method available to date for quantifying full capsid ratio in crude lysate

samples, and it is commonly used by many laboratories across industry and academia (e.g., PMID: 35256603, 37096037, 39941089). Nonetheless, for a representative comparison between AAVPure^{Mfg} and triple transfection, we tried to set up affinity chromatography purification (as our established density gradient purification method separates full from empty capsids). Additionally, we were collaborating with a colleague to establish an AUC protocol. Besides the technical difficulties, we were also facing nationwide research budget cuts and recently imposed institutional spending constraints, which further complicated the resource-intensive tasks of large-scale rAAV production, chromatography purification, and AUC characterization. Therefore, despite our best efforts, we were unable to provide AUC data to address the limitation associated with the ddPCR/ELISA method. At the editor's discretion, we provide our detailed explanation as outlined above, and have discussed this limitation in the revision (**lines 351-357**).

- Testing for rcAAV and comparing against standard triple transfection

Response 3-1b: We performed single-molecule, long-read DNA sequencing (PacBio) of encapsidated vector DNA to further characterize DNA impurities. Regarding replication-competent AAV (rcAAV), we detected a single read of Rep-Cap out of approximately 600,000 reads in each sample, suggesting low and comparable levels of potential rcAAV across both platforms. These data are included in the revision (**Table EV2, lines 189-191**).

- Comparing rescue out of and subsequent genome replication between standard ITR plasmid format and AAVPUREMFG plasmid format (minicircle).

Response 3-1c: We thank the reviewer for the suggestion to compare vector genome rescue and replication. We extracted total cellular DNA from cells undergoing either triple transfection or AAVPure^{Mfg} 2.0, followed by DpnI treatment to remove residual plasmids before vector genome quantification. This is a well-established procedure to quantify viral DNA in the presence of interfering plasmid DNA (e.g., PMID: 19211760). ddPCR detected similar levels of DpnI-resistant vector genome DNA between the two platforms, suggesting comparable rescue/replication efficiency between standard ITR plasmid format and AAVPure^{Mfg} plasmid format (minicircle). This set of data have been included in the revision (**Appendix Figure S5, lines 245-248**).

Minor comment #2:

For AAVPureMfg 3.0 stable cell line, reviewer recommends testing for residual Puromycin

Response 3-2: We agree and have designed a high-sensitivity, targeted ddPCR assay to quantify potential Puromycin contaminants in purified AAV vectors produced with AAVPure^{Mfg} 3.0. This analysis showed that any possible Puromycin gene encapsidation would be below the detection limit of ddPCR. These data have been included in the revision (**Appendix Figure S7, lines 340-341**).

Minor Comment #3:

Transfection DoEs could be performed to further improve yields and F:P:E packaging (see comment 1) using the AAVPureMfg plasmids.

Response 3-3: We agree with the reviewer that a DoE approach could be used to systematically optimize transfection variables for higher rAAV production yield and quality. As a proof-of-concept, we screened multiple conditions in the new experiments to package a large transgene or a sc genome, with a focus on the two key plasmids used in AAVPure^{Mfg} (**Figure EV3C-F**). Although it was not a comprehensive DoE approach, the data demonstrated the potential of optimizing transfection conditions to maximize yield. Additionally, it suggested that effective optimization, such as using DoE,

could be performed in the context of specific transgene and genome configuration. While we respectfully believe that conducting a systematic DoE experiment with the EGFP model construct would be beyond the scope of this study, we have discussed the potential integration of a DoE approach into AAVPure^{Mfg} (lines 348-351).

Minor Comment #4:

Reviewer acknowledges that the experiments performed at 30 mL scale in suspension Expi293 cells will scale to larger volumes based on transfection scalability shown by several gene therapy companies and CMOs. It would be valuable in this manuscript or a future manuscript using AAVPureMfg to perform two 3L to 10L bioreactor manufacturing runs with an industry standard downstream purification process (chromatography with or without iodixanol density gradient) to generate enough rAAV vector material to perform standard QC release testing (including residuals, infectivity/potency, rcAAV) to compare against triple plasmid transfection.

Response 3-4: We fully agree with the reviewer that a comparative study with industry-relevant production scale, upstream process, downstream process, and analytical release testing would strengthen the utility of AAVPure^{Mfg}. We are actively seeking collaboration with an industry partner and will conduct such a study when opportunities arise. This direction is added in the revision (lines 356-357).

Overall, the manuscript is technically sound, the experiments conducted along with their respective figures are detailed and graphically presented well and the conclusions reached based on the data are accurate. The reviewer's comments are minor and for the authors to consider for future experiments as they continue to validate and improve the AAVPUREMFG technology. The reviewer recommends accepting this manuscript for publication as the authors have demonstrated POC that will ultimately improve rAAV vector manufacturing.

Response 3-5: We thank the reviewer for the thoughtful comments and suggestions on the future development of this technology.

25th Apr 2025

Dear Dr. Wang,

We are pleased to inform you that your manuscript is accepted for publication and is now being sent to our publisher to be included in the next available issue of EMBO Molecular Medicine.
